# *RuAG*: Learned-rule-augmented Generation for Large Language Models

**Yudi Zhang[1]\*, Pei Xiao[2]\*, Lu Wang[3]†, Chaoyun Zhang[3], Meng Fang[4]† , Yali Du[5],**
**Yevgeniy Puzyrev[3], Randolph Yao[3], Si Qin[3], Qingwei Lin[3], Mykola Pechenizkiy[1],**
**Dongmei Zhang[3], Saravan Rajmohan[3], Qi Zhang[3]**

[1]Eindhoven University of Technology    [2]Peking University    [3]Microsoft
[4]University of Liverpool    [5]King's College London

## Abstract

In-context learning (ICL) and Retrieval-Augmented Generation (RAG) have gained attention for their ability to enhance LLMs' reasoning by incorporating external knowledge but suffer from limited contextual window size, leading to insufficient information injection. To this end, we propose a novel framework *RuAG* to automatically distill large volumes of offline data into interpretable first-order logic rules, which are injected into LLMs to boost their reasoning capabilities. Our method begins by formulating the search process relying on LLMs' commonsense, where LLMs automatically define head and body predicates. Then, *RuAG* applies Monte Carlo Tree Search (MCTS) to address the combinational searching space and efficiently discover logic rules from data. The resulting logic rules are translated into natural language, allowing targeted knowledge injection and seamless integration into LLM prompts for LLM's downstream task reasoning. We evaluate our framework on public and private industrial tasks, including natural language processing, time-series, decision-making, and industrial tasks, demonstrating its effectiveness in enhancing LLM's capability over diverse tasks. Project link: `https://github.com/microsoft/RuAG`.

## 1 Introduction

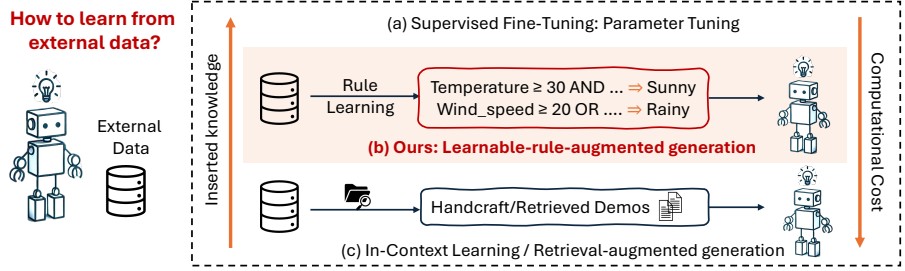

Figure 1: Comparison of supervised fine-tuning, in-context learning/retrieval-augmented generation, and our proposed learned-rule-augmented generation (*RuAG*), which injects logic knowledge to boost generation while reducing computational cost.

Leveraging external datasets to enhance the performance of pretrained Large Language Models (LLMs) on downstream tasks has become a significant focus in recent research (Brown et al., 2020a; Hu et al.; Fan et al., 2024; Dong et al., 2022). Methods such as supervised fine-tuning (SFT) (Hu et al., 2021; Li & Liang, 2021), in-context learning (ICL) (Dong et al., 2022; Wang et al., 2020; Ravi & Larochelle, 2016; Chan et al., 2022; Fang et al., 2024), retrieval-augmented generation

---

*Equal contribution. Work done during the internship of Yudi and Pei in Microsoft.
†Correspondence to: Lu Wang <wlu@microsoft.com>, Meng Fang <Meng.Fang@liverpool.ac.uk>.

(RAG) (Izacard et al., 2023; Fan et al., 2024), and the utilization of knowledge graphs (KGs) (Pan et al., 2024; Shu et al., 2024; Wang et al., 2024) have been explored to incorporate external knowledge into LLMs (Ding et al., 2023; Zhang et al., 2024), enhancing their reasoning and decision-making capabilities.

Despite these advancements, these methods face notable challenges. Fine-tuning large LLMs on extensive datasets is computationally intensive and time-consuming, often leading to overfitting and catastrophic forgetting (McCloskey & Cohen, 1989). ICL relies on handcrafted demonstrations and templates that may not effectively summarize large volumes of data, leading to inefficiencies and the "needle in a haystack" problem when processing long contexts (Li et al., 2024), and the extremely long context window significantly increases computational costs (Peng et al., 2024; Naveed et al., 2023). RAG depends heavily on the quality and relevance of retrieved documents and faces computational hurdles when integrating large-scale retrieval into prompts (Fan et al., 2024). Thus, RAG is not able to use the whole of vast knowledge base. KG-based methods incorporate structured representations of knowledge to improve LLMs' understanding and reasoning (Pan et al., 2024; Shu et al., 2024; Wang et al., 2024). While KGs can enhance decision-making by providing explicit relational data, constructing and maintaining them requires significant manual effort and domain expertise, making scalability challenging.

These challenges underscore the urgent need for efficient knowledge transformation to enhance LLMs' understanding. Logic rules, with their high information density, act as a promising bridge between vast, diverse data types (including numerical, textual, and visual data) and LLMs' understanding. Previous

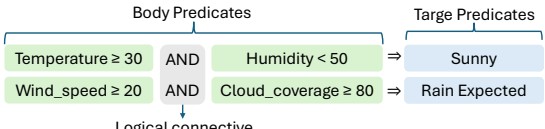

Figure 2: Illustration of logic rules, which are utilized to guide LLMs' generation in this paper.

work has demonstrated their learnability from external data and their efficiency in providing explanations to enable transparent AI processes (Qu & Tang, 2019; Qu et al.). A logic rule, as shown in Figure 2, typically expressed as $\alpha \to h$, indicates that if a set of events $\alpha$ (referred to as body predicates) occurs, then the event $h$ (called the target predicate) will also occur. As an example, the logic rule "Temperature $\geq 30$ AND Humidity $\leq 50 \to$ Sunny Day" represents knowledge in symbolic structures, suitable for learning from data. Additionally, this rule can be easily translated into natural language: "If the temperature is 30 degrees or higher and the humidity is 50 percent or lower, it will be a sunny day." Logic rules are understandable to both humans and LLMs as they encapsulate complex relationships in a concise, structured form. Unlike lengthy text passages or extensive datasets in ICL and RAG, logic rules distill essential information into clear, interpretable statements. Compared to the complex node-and-edge structure of KGs, logic rules reduce cognitive load and align better with LLMs' natural language training. Their direct translation into natural language further improves alignment with LLMs, facilitating more efficient processing and understanding.

Inspired by this, we propose a novel framework, **learned-rule-augmented generation (*RuAG*)**, to automatically compress large external data into logic rules through LLM-aided Monte Carlo Tree Search (MCTS) (Świechowski et al., 2023) and then inform LLMs domain expertise by applying translated logic rules into prompts. Our framework consists of the following three phases. **LLM-based Logic Rule Search Formulation:** Learning logic rules is expensive due to the involved human effort in formulating the domain-specific search process. Therefore, we automate this process by relying on LLMs' commonsense to define the target and body predicates in logic rules. First, the target predicate is defined to be task-relevant, like a class label in a classification task or a game state labeled as "win", while the body predicates are initialized as all the data attributions in the dataset. Then, given the task and dataset descriptions, LLM generates new target predicates and eliminates most irrelevant data attributions from the body predicates. For example, in navigation, LLMs may infer some special place as the key steps to the destination and suggest to search the rules for agents reaching the places individually. Also, LLMs may regard some data attributes as irrelevant to the target predicate, thus excluding them from the candidates. Consequently, the logic rule search space can be significantly reduced, and a domain-specific search process can be automatically established. **Logic Rule Search with MCTS:** Searching rules requires to discover the relationship among the predicates, suffering from the compositional search space (Qu & Tang, 2019; Zhang et al., 2020; Evans & Grefenstette, 2018). To this end, *RuAG* exploits MCTS, which works well in large search spaces, to generate structured and understandable first-order logic rules, which are applied in the

rule-based generation phase. **Learned-Rule-Augmented Generation:** *RuAG* translates the abstract logic rules into natural language and injects them into LLMs' prompts. By addressing the limitations of SFT, ICL, RAG, and KG-based methods, *RuAG* offers a scalable and computationally efficient solution for integrating extensive domain knowledge into LLMs, improving LLM's reasoning, comprehension, and task performance with minimal manual intervention.

Our contributions are fourfold. First, we introduce a novel learned-rule-augmented generation framework as a potential alternative to SFT, ICL, RAG, and KG-based methods. This framework systematically and nearly automatically compresses external knowledge into compact, interpretable logic rules that prioritize enhancing LLM generation. Second, we propose an automated formulation for MCTS, eliminating the need for manual, domain-specific rule search and enabling a generalizable approach applicable across a wide range of tasks. Third, we apply MCTS to efficiently handle the large compositional search space of logic rule discovery. Fourth, we evaluate our framework across diverse scenarios, including public tasks in NLP (relation extraction on DWIE), time-series (log anomaly detection on HDFS), decision-making (the cooperative game Alice and Bob), and an industrial task in abuse detection, demonstrating the effectiveness of our approach in both academic and real-world settings.

## 2 RELATED WORK

In this section, we review the most relevant topics related to our work, including the techniques to exploit external data in LLMs and logic rule learning.

**External data usage in LLMs.** There are several ways to inject external knowledge into large language models. The most common way is distilling knowledge from external data, but it suffers from high computational costs (Xu et al., 2024; Gu et al.; Zhang et al., 2025). In-context learning (Brown et al., 2020a) prompts LLMs with a few handcrafted demonstrations which are understandable for the LLMs. More fancy, Retrieval-Augmented Generation (RAG)(Chen et al., 2024a) complements LLMs by retrieved relevant knowledge from external databases (Li et al., 2023; Shen et al., 2023; Chen et al., 2024c) or constructing demonstrations for in-context learning (ICL) (Poesia et al., 2022; Agrawal et al., 2023), showing promise in tasks like OpenQA (Borgeaud et al., 2022; Guu et al., 2020) and games (Zhu et al., 2023a; Hu et al., 2024). Knowledge graphs are welcome in external knowledge formats as well, especially in structured tasks like relation extraction and entity recognition (Shu et al., 2024; Wang et al., 2024), improving task-specific decisions. Recent research also investigates how LLMs can summarize logic rules from large datasets to serve as a knowledge storage (Zhu et al., 2023b; Luo et al., 2023), but shows high computational costs due to frequent calls to commercial LLMs (Brown et al., 2020b; OpenAI, 2023).

**Logic rule learning.** Logic rules are increasingly employed to enhance the interpretability and accuracy of decision-making in AI systems (Chiu et al., 2023; An et al., 2024). Manually defined logic rules have been used to describe how certain events or outcomes are triggered by predefined conditions. However, this process is labor-intensive and highly domain-dependent (Evans & Grefenstette, 2018; Li et al., 2020). Researchers have explored automatic methods for extracting logic rules, such as statistical approaches and likelihood estimation (Cheng et al., 2022; Qu et al.; Ru et al., 2021). Despite these advances, the process still involves extensive domain knowledge and commonsense reasoning, requiring expert intervention to identify the candidate target and body predicates.

## 3 ENHANCE LLMS' REASONING THROUGH APPLYING LOGIC RULES

In this section, we introduce *RuAG*, our novel approach to augment Large Language Models (LLMs) with logic rules learned from pre-collected training data. Instead of directly fine-tuning the LLM—which can be costly and prone to overfitting—or using retrieval-augmented generation limited by input length, we transform the data into concise logic rules. These rules encapsulate essential patterns and guide the LLM during generation, enhancing performance and interpretability.

As shown in Figure 3, *RuAG* comprises three key steps: 1) **LLM-Based Logic Rule Search Formulation:** leverage the LLM to automatically formulate the *logic rule learning problem*, defining predicates, actions, states, and rewards. (Section 3.1) 2) **Logic Rule Search with Monte Carlo Tree Search (MCTS):** employ MCTS to efficiently search for effective logic rules based on the

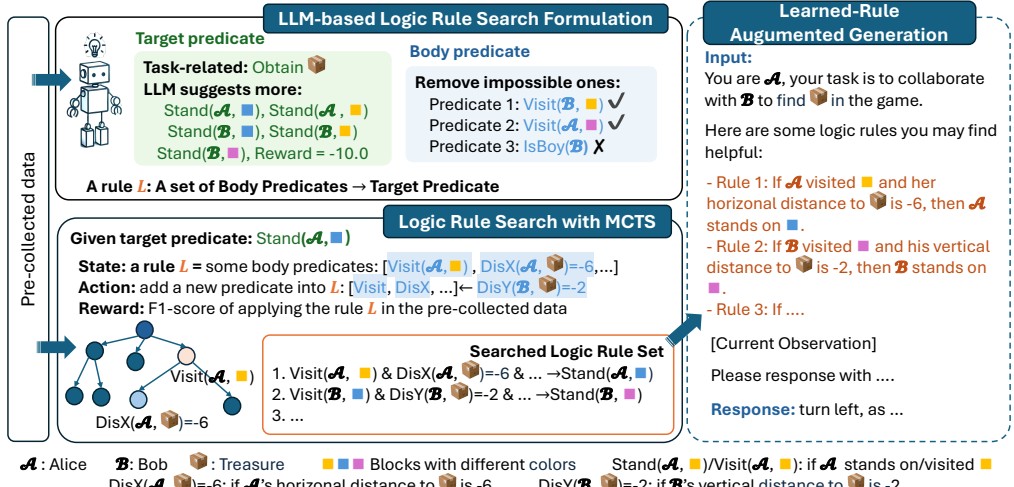

Figure 3: The framework of our novel learned-rule-augmented generation (*RuAG*). *RuAG* automatically compresses large external knowledge into compact logic rules using LLM-aided Monte Carlo Tree Search (MCTS), through three phases: **LLM-based Logic Rule Search Formulation**, **Logic Rule Search with MCTS**, and **Learned-Rule-Augmented Generation**. First, the LLM formulates the MCTS search by defining the target and body predicates. Then we apply MCTS to generate structured first-order logic rules, which are applied to guide generation. Our framework provides an efficient alternative to RAG.

Table 1: Task-relevant target predicates, body candidates and their examples in relation extraction, log-based anomaly detection and cooperative game.

| Task | Training data | Target predicate | Body predicate candidates |
|------|---------------|------------------|---------------------------|
| Relation Extraction | relations between entities, e.g., `in0(A,C)` (A is in country C) | A specific relation. | All relations except the target predicate, e.g., for target predicate `in0(A,C)`, all other relations excepting `in0(A,C)`. |
| Anomaly Detection | Pairs of log sequences and anomaly labels (`Anomaly`) | `Anomaly` | Log events, e.g., `E5` (receiving a block). |
| Cooperative Game | Triplets of observations, actions and game win label (`GameWin`) | `GameWin` | Observations and actions, e.g., `IsYellow(Alice, Right)`: Alice's right block is yellow, `Move(Bob, Right)`: Bob moves right. |

LLM-formulated problem. (Section 3.2) 3) **Learned-Rule-Augmented Generation:** integrate the learned logic rules into the LLM's generation process, improving its generation. (Section 3.3)

## 3.1 FROM DATA TO RULE SEARCH: LLM-BASED LOGIC RULE SEARCH FORMULATION

Search for logical rules traditionally requires significant human effort, particularly in defining domain-specific head predicates and selecting relevant features that characterize data samples. This process demands domain knowledge and impacts both the quality of the derived logic rules and the computational cost of search. To address this challenge, our method begin with LLM-based Logic Rule Search Formulation, where we leverage the capabilities of LLMs to automatically formulate the logic rule learning problem through defining the predicates.

**Initial Predicates.** Given a dataset $\mathcal{D} = \{(\boldsymbol{x}, y)\}$, where each data sample $\boldsymbol{x} = [x_1, x_2, \ldots, x_N] \in \mathcal{X}$ is $N$-dimensional and $y \in \{0, 1\}$ is the label, we initial the label as the target predicate and the features as the body predicates, as shown in Table 1. We can directly translate discrete variables into Boolean values through one-hot vectors. For continuous variables, we can translate them into Boolean-valued attributes through Gini-index (Strobl et al., 2007). Furthermore, we suggest prompting LLMs to remove impossible body predicates to reduce logic rules search space or suggest new target predicates to search more logic rules for a better understanding of the task.

**Removing Impossible Body Predicates.** Given the definition of logic rules and the candidate predicates with their descriptions, LLM aids in filtering out impossible or irrelevant body predicates, reducing the computational burden. By utilizing commonsense reasoning, the LLM can identify

predicates that are unlikely to contribute to effective logic rules. For instance, in a system log analysis, the LLM might determine that certain attributes like user IDs are less relevant for anomaly detection compared to error codes or access patterns. We provide the prompt in Figure A3.

**Suggesting New Target Predicates.** In addition to the primary target predicate (e.g., achieving a specific classification label), the LLM can suggest additional head predicates to explore, given logical rule definition, task description, and data schema. This is particularly useful in tasks requiring long-horizon planning, where intermediate goals can guide the search for effective logic rules. By generating these new head predicates, the LLM enables a more comprehensive exploration of the logic rule space. We provide the prompt in Figure A4.

Our LLM-based logic rule search formulation enjoys the following advantages:

- **Automation and Scalability:** The LLM automates the setup of the logic rule learning problem, *i.e.*, defining the target and body predicates, avoiding human experts, and making it scalable to large and complex datasets.
- **Enriched rule generation:** By generating relevant target predicates, our method can extract more meaningful rules.
- **Reduced Computational Burden:** By eliminating irrelevant predicates, the LLM narrows down the search space, improving efficiency.

### 3.2 LOGIC RULE SEARCH WITH MCTS

Following the definition of predicates in logic rule searching, we apply Monte Carlo Tree Searching (MCTS) to perform logic rule learning, inspired by its effectiveness in searching optimal policy in large state spaces.

**States, Actions, and Rewards in MCTS.** With the predicates defined, the state, action, and reward in MCTS for logic rule searching can be defined as:

- **States** ($S$)**:** Each state represents a partial logic rule, consisting of a set of predicates. The initial state is the empty set, $S_0 = \emptyset$. Subsequent states are defined as: $S_n = S_{n-1} \cup \{\alpha_i\}$, where $\alpha_i$ is the predicate added by an action.
- **Actions** ($\mathcal{A}$)**:** Actions involve adding a new predicate to the current state. The action space is defined as: $\mathcal{A} = \{\text{Add } \alpha_i \mid \alpha_i \text{ is a candidate predicate generated by the LLM}\}$.
- **Rewards** ($\mathcal{R}$)**:** The reward function evaluates the quality of a logic rule. For example, the reward for state $S_n$ can be defined as the precision of the rule evaluating on the dataset $\mathcal{D}$.

Typically, MCTS involves building a search tree and simulating outcomes to estimate the value of actions. It consists of four key phases: selection, expansion, simulation, and backpropagation. **Selection and expansion.** The process begins at the root node, where the algorithm selects the most promising child nodes based on the Upper Confidence Bound applied to Trees (UCT). This continues until a leaf node is reached. If the leaf node is not terminal, new child nodes are created to explore potential moves. As an example, we expand a new node at the state of $[(\text{age} \geq 30),] \Rightarrow (\text{income} \geq \$50,000)$, if we select a new candidate predicate $(\text{income} \geq \$50,000)$ according to its UCT value, then we add it into the rule: $[(\text{age} \geq 30),] \leftarrow (\text{income} \geq \$50,000)$ and enter the new state of $[(\text{age} \geq 30), (\text{education} = \text{bachelor's})] \Rightarrow \text{income} \geq \$50,000$. **Simulation**: For the newly expanded nodes, random simulations (also known as rollouts) are performed to calculate the reward of the state. **Backpropagation:** The calculated reward is then propagated back up the tree, updating the nodes' statistical information. The UCT algorithm plays a crucial role in MCTS, balancing exploration and exploitation by selecting actions that maximize: $UCT_j = \bar{X}_j + C\sqrt{\frac{2 \ln N_C}{N_j}}$, where $\bar{X}_j$ is the average reward of action $j$, $N_C$ is the total number of visits to the parent node, is the number of visits to node $j$, $C$ is a constant that adjusts the exploration-exploitation trade-off.

Finally, we collect all the rules constructed at the terminal nodes when 1) the constructed rule reaches a predefined maximum length (i.e., the number of body predicates exceeds a threshold).2) If the reward of the final node (i.e., the precision of the rule) exceeds a predefined threshold, indicating that the rule is sufficiently accurate.

### 3.3 LEARNED-RULE-AUGMENTED GENERATION

After the logic rule search, we gather a set of logic rules and follow the following steps to perform learned-rule-augmented generation. 1) **Clean Searched Rules:** The collected rules may contain

duplicates, exhibit low quality, or cover only a limited subset of the data. We first eliminate those with low rewards or minimal data coverage. Then, we compare each pair of rules and retain the one with the higher reward if its body predicates are a subset of the other's. 2) **Translate Rules into Natural Language:** To enhance the LLMs' comprehension, we translate these symbolic rules into natural language, resulting in a group of sentences. These sentences can then be injected into the LLM prompts to guide generation more effectively. 3) **Retrieve Relevant Rules:** It is optional to retrieve only the most relevant rules or inject all the rules, depending on the contextual window size and the long-text understanding capability of the LLM. 4) **Generation:** The generator component can be modeled using any LLM. We use GPT-4 (OpenAI, 2023) if no specific model is clarified. To combine the input with the rules during generation, we simply apply the rules in a prompt template.

## 4 EXPERIMENTS

Most decision-making and prediction tasks can be abstracted into state chains to achieve their ultimate goals, which allows our method to adapt to a wide variety of tasks. In this section, we evaluate our method over diverse domains, including NLP (relationship extraction in Section 4.1), time-series predication (log-based anomaly detection in Section 4.2), decision-making task (cooperative game (Chen et al., 2024b) in Section 4.3) and a private industrial task (unauthorized party abuse detection in Appendix A). We compare our method with the domain-specific baselines for each task and LLM-based methods, including vanilla, ICL, RAG, and HtT (Zhu et al., 2023b). HtT shares a similar goal of using constructed rule libraries with us to enhance LLM generation., but relies on LLMs to generate and verify rules, leading to high computational cost of LLMs. The specific implementation details of the experimental setup can be found in Appendix C.

### 4.1 RELATION EXTRACTION

Document-level relation extraction is a critical task in natural language processing (NLP), where the goal is to identify and classify relationships between entities across entire documents rather than isolated sentences. This task becomes more complex at the document level due to the larger context and the need to resolve long-range dependencies and co-references between entities scattered throughout the document. However, using only LLMs for this task is often limited by their inability to consistently capture complex document-wide relationships, especially when reasoning across multiple entities and contexts.

**Setup.** We conduct experiments on the DWIE dataset (Zaporojets et al., 2021), which contains 802 documents and 23,130 entities. After excluding irrelevant articles, 700 documents are used for training and 97 for testing. we utilized the LLM to identify and eliminate 15% of the relationships (i.e. appears_in,vs and player_of) that were unlikely to function as valid body predicates, based on their descriptions and the given rule descriptions. We evaluate the performance of our method using standard relation extraction metrics, including Precision, Recall, and F1-score. For comparison, we evaluate our method against several state-of-the-art models for document-level relation extraction, including CNN, BiLSTM (Yao et al., 2019), Context-Aware (Sorokin & Gurevych, 2017), and BERT-based models(Shi & Lin, 2019), which are widely used in document-level relation extraction tasks. Additionally, we compare with the LLM-based methods: Vanilla, ICL, RAG and HtT(Zhu et al., 2023b) which employs predefined logical rules to extract relations.

**Main Results.** As shown in Figure 2, our method outperforms both deep learning-based and LLM-based baselines in document-level relation extraction. Deep learning methods, while achieving decent performance in document-level relation extraction, struggle to capture long-range semantic dependencies. LLMs, such as GPT-4, demonstrate superior performance due to their strong natural language understanding. For example, ICL achieves an F1 score of 50.26%, outperforming DL-based methods. RAG further improves results (F1:

Table 2: Results on Relation Extraction.

|  | Model | F1 | Precision | Recall |
|---|---|---|---|---|
| DL-based | CNN | 43.78% | 47.13% | 45.03% |
|  | BiLSTM | 48.17% | 44.32% | 41.53% |
|  | Bert | 49.84% | 49.35% | 54.13% |
|  | Context-Aware | 45.37% | 49.87% | 38.76% |
| LLM-based | Vanilla | 46.94% | 69.61% | 35.41% |
|  | ICL | 50.26% | 74.09% | 38.02% |
|  | RAG | 52.30% | 78.64% | 39.17% |
|  | HtT | 52.59% | 68.20% | 42.80% |
|  | **Ours** | 60.42% | 69.44% | 53.48% |

52.30%) by retrieving similar cases from a training-based knowledge base, while HtT extracts rules document by document (F1: 52.59%), which limits its global perspective. Our method addresses these limitations by using MCTS to search for rules globally, mining potential rules efficiently from the entire training data. This ensures reliability during the search process and combines the learned rules with LLM reasoning capabilities, achieving an F1 score of 60.42%. These results highlight the effectiveness of our approach in delivering more accurate and comprehensive relation identification compared to both traditional DL-based and other LLM-based methods.

## 4.2 LOG-BASED ANOMALY DETECTION

Log-based anomaly detection is fundamentally a time-series prediction task, where the goal is to predict whether a sequence of log events indicates abnormal system behavior. This task is crucial for maintaining system reliability and security by identifying patterns that signal potential failures or attacks. Given the temporal nature of log data, both sequential patterns and the semantic content of the logs must be analyzed to accurately detect anomalies. Effective anomaly detection in time-series log data is essential for preventing system downtime and ensuring the smooth functioning of distributed infrastructures.

**Setup.** We evaluate our method on the HDFS dataset (Xu et al., 2009) for the log-based anomaly detection task. This dataset consists of over 11 million log entries generated from Hadoop-based map-reduce jobs on more than 200 Amazon EC2 nodes. In practice, we sampled 20,000 blocks of log sequences from the HDFS dataset, consisting of approximately 486,060 log entries. The dataset is split chronologically into training, validation, and test sets with a ratio of 8:1:1. We evaluate our method using F1 score, Precision, and Recall to compare it against several baselines, including traditional methods like LogCluster (Lin et al., 2016), DeepLog (Du et al., 2017), and LogRobust (Zhang et al., 2019), as well as LLM-based models like Vanilla, HtT, LogGPT (Qi et al., 2023), ICL and RAG, providing a comprehensive assessment of performance across various approaches.

**Main Results.** Table 3 compares our method with traditional baselines and LLM-based models on the log-based anomaly detection task. Traditional deep learning methods like LogCluster and DeepLog rely heavily on training data, making it hard for them to detect new anomalies. While LogRobust improves performance (F1: 87.31%), its ability to generalize remains limited. LLMs like GPT-4 perform well using simple prompts. For example, LogGPT achieves an F1 score of 72.56%, showing strong semantic reasoning. However, its low precision (56.82%) leads to misclassifications of mi-

Table 3: Comparison under different methods on Log-based anomaly detection.

|  | Mehtod | F1 | Precision | Recall |
|---|---|---|---|---|
| DL-based | LogCluster | 70.97% | 96.70% | 56.05% |
|  | DeepLog | 79.64% | 84.81% | 75.08% |
|  | LogRobust | 87.31% | 89.12% | 85.54% |
| LLM-Based | Vanilla | 60.10% | 47.05% | 83.16% |
|  | ICL | 69.77% | 78.95% | 62.50% |
|  | RAG | 84.32% | 98.97% | 73.46% |
|  | LogGPT | 72.56% | 56.82% | 100% |
|  | HtT | 58.73% | 45.46% | 82.31% |
|  | **Ours** | **92.59** | **100%** | **86.21%** |

nor issues as anomalies. HtT learns patterns from training data but struggles with efficiency and global pattern recognition, resulting in an F1 score of 58.73%. RAG improves LLM performance by retrieving similar cases, but the limited information in a single case restricts its effectiveness. Our method addresses these issues by using MCTS to extract reliable rules from the entire dataset, providing clear guidance to the LLM. This approach eliminates misclassifications, achieving an F1 score of 92.59%, outperforming all baselines and effectively balancing accuracy and generalization.

## 4.3 MULTI-AGENT GAME: ALICE&BOB

In the real world, plenty of scenarios involve decision-making, planning, and collaborating, especially in partially observable environments. Moreover, often the optimal strategy often contradicts human intuition. You can not walk towards the treasure directly as there may be walls blocking the path. In such tasks, it is crucial to inject domain knowledge to make informed decisions, as only by integrating specific domain expertise can the model accurately identify the optimal strategy and make sound judgments.

**Setup.** We choose the cooperative multi-agent game *Alice&Bob*, which requires both planning and collaboration. In the game, Alice and Bob work together to find the treasure (Chen et al., 2024b),

and the optimal paths for both agents often go against intuition. They are required to sequentially experience key blocks, with one agent needing to remain on a block to enable the other to obtain the treasure. Episodes last up to 50 steps. **Metric** We evaluate the method by reporting the average win rate (**WR**), the accumulative rewards (**AR**), and the average episode length (**AL** across 30 episodes. **Baselines.** We compare our method with RL baselines (behavior cloning; offline tabular Q), rule generated method (PLLB (Srivastava et al., 2024) and HtT (Zhu et al., 2023b)), ICL-Good (ICL with 3 good demonstrations) and ICL-Contrastive (ICL with 2 good and 2 bad demonstrations) and RAG, RAG retrieves timesteps with similar observations and informs LLMs both observations and actions. We provide the results of random policy and LLM-based grounded policy (with handcraft rules) as well. **Data Collection** We collect 1000 episodes of trajectories by applying a handcraft policy where the agent has the probability of $p$ to follow the optimal policy and $1 - p$ to follow a random policy. We set $p = 0.7$ in the default setting. **Generated target predicates by LLMs.** We search the logic rules from different aspects following the LLMs' suggestion: 1) team reward = -10; 2) Alice or Bob stand on yellow, purple, skyblue blocks; 3) Game Win. During the evaluation, we make different LLM serves as Alice and Bob, providing them with the observations, historical information, and the action space and prompting them to respond with the chosen action.

**Main Results.** In Table 4, we compare the performance of various RL-based and LLM-based methods on the Alice & Bob task. Overall, our method achieves the sota performance. RL-based methods perform relatively well and surpass most LLM-based methods, as they can accumulate knowledge during training. In contrast, LLM-based methods face significant challenges in this task. Methods like Vanilla, ICL-Good, and ICL-Contrastive show negative accumulative rewards (-0.08, -0.71, and -0.83, respectively) with a win rate of 0, indicating a

Table 4: Experimental results over the decision-making task, *Alice&Bob*. The standard error is provided in the bracket.

|  | Method | AR | AL | WR (%) |
|---|---|---|---|---|
| RL-based | Behavior Cloning | 54.67($\pm$51.82) | 32.46 | 0.56 |
|  | Offline Tabular Q | 59.51($\pm$52.71) | 32.60 | 0.63 |
| LLM-based | Vanilla | -0.08($\pm$0.11) | 50.0 | 0.0 |
|  | ICL-Good | -0.71($\pm$0.55) | 50.0 | 0.0 |
|  | ICL-Contrastive | -0.83($\pm$0.66) | 50.0 | 0.0 |
|  | RAG | -0.14($\pm$0.22) | 50.0 | 0.0 |
|  | HtT | -0.26 ($\pm$0.22) | 50.0 | 0.0 |
|  | PLLB (Offline) | -0.15($\pm$0.26) | 50.0 | 0.0 |
|  | **Ours** | **69.45($\pm$46.1)** | 33.23 | 0.7 |
|  | Random | -2.2($\pm$0.52) | 50.0 | 0.0 |
|  | Grounded | 89.87($\pm$30.06) | 32.1 | 0.9 |

clear lack of strategy reasoning and task optimization. Vanilla performs badly due to the absence of domain knowledge. However, once domain knowledge is correctly incorporated, performance improves significantly, as seen with methods like Ours (win rate of 0.7) and Grounded Policy (win rate of 0.9). Among those in-context-based LLM methods, ICL and RAG insert relevant demonstrations. However, they perform bad as LLMs may suffer from long-text understanding. HtT, and PLLB rely on LLM to summarize rules, which not only need to understand long text but also require more domain knowledge than our method to summarizing rules, therefore the summarized rules may not provide enough domain knowledge for LLMs.

## 4.4 ABLATION STUDY

In this section, we present an ablation study to evaluate the robustness and effectiveness of our method across several dimensions. First, we analyze the performance of our method when using different LLM backbones, examining whether the choice of LLM impacts overall task performance. Second, we explore the contribution of different components in our method, including the use of chain-of-thought (CoT) reasoning and rule-based guidance, to assess how each component improves the task. Lastly, we investigate the effectiveness of the MCTS rule extraction process by varying the number of search episodes.

**Ablation on Different LLM backbones.** Table 5 presents the results of our ablation study on different LLM backbones across relation extraction, log anomaly detection and cooperative games. It compares baseline models (Vanilla), chain-of-thought (CoT), and our *RuAG* for GPT-3.5 and GPT-4. While CoT improves performance by promoting step-by-step reasoning, it falls short in tasks requiring domain knowledge. In contrast, *RuAG* learned rules from external data, provides the required context, and consistently enhances performance across different backbones.

**Ablation on searching episodes in MCTS.** Table 6 shows the impact of MCTS search episodes for three tasks. In relation extraction and cooperative games, we report the number and accuracy of

Table 5: Ablation on LLM backbones across different tasks.

| Backbone | Method | Relation Extraction | | | Log Anomaly Detection | | | Cooperative Game | | |
|---|---|---|---|---|---|---|---|---|---|---|
| | | F1 | Precision | Recall | F1 | Precision | Recall | AR | AL | WR |
| GPT3.5 | Vanilla | 18.94% | 31.06% | 13.62% | 48.42% | 62.71% | 39.43% | -0.58(±0.47) | 50.0 | 0.0 |
| | +CoT | 19.85% | 28.19% | 15.32% | 73.19% | 75.42% | 71.08% | -0.38(±0.26) | 50.0 | 0.0 |
| | +rule | 26.63% | 39.82% | 20.00% | 91.39% | 100.00% | 84.16% | 45.2(±49.81) | 42.73 | 0.45 |
| GPT4 | Vanilla | 46.94% | 69.61% | 35.41% | 60.10% | 47.05% | 83.16% | -0.08(±0.11) | 50.0 | 0.0 |
| | +CoT | 48.10% | 66.13% | 37.39% | 76.11% | 63.62% | 94.69% | -0.83(±0.66) | 50.0 | 0.0 |
| | +rule | 60.42% | 69.44% | 53.48% | 92.59% | 100.00% | 86.21% | 69.45(±46.1) | 33.23 | 0.7 |

Table 6: Ablation on searching episodes in MCTS. Num. denotes the number of searched rules.

| Times | Relation Extraction | | Anomaly Detection | | | Cooperative Game | |
|---|---|---|---|---|---|---|---|
| | Num. | Precision | F1 | Precision | Recall | Num. | Precision |
| 50 | 13 | 100% | 65.75% | 100.00% | 48.89% | 14 | 100% |
| 200 | 20 | 100% | 86.86% | 98.7% | 77.55% | 16 | 100% |
| 500 | 21 | 100% | 91.30% | 100% | 84% | 21 | 100% |
| 1000 | 23 | 95.65% | 91.30% | 100% | 84% | 23 | 91.30% |

extracted rules are evaluated, while log anomaly detection is assessed based on the final task performance. According to the results, fewer search episodes still yield high-quality rules. Increasing episodes expands the search space, leading to more rules, but with diminishing returns as excessive episodes introduce ineffective searches and, in some cases, incorrect rules (e.g., relation extraction).

**Ablation on hyperparameter $p$ for data collection in decision-making task.** We adjust the probability $p$ of performing optimal policy and report the searched rule numbers and their precision in Table 7 to investigate the impact of data collection policies on the searched rules.

## 4.5 CASE STUDY

In this section, we present a case study to demonstrate how the extracted rules help LLMs perform tasks more effectively across different domains. The extracted rules serve as a guiding mechanism, assisting the LLM in making more accurate predictions and improving task performance by providing structured logic and patterns that the LLM can follow. Figure 4 illustrates the most representative cases where extracted rules helped LLMs improve performance across three tasks: relation extraction, log-based anomaly detection, and multi-agent gaming.

Table 7: Ablation on hyperparameter $p$.

| $p$ | Num | Precision |
|---|---|---|
| 0.2 | 25 | 80% |
| 0.5 | 35 | 88% |
| 0.7 | 21 | 100% |

In the relation extraction task, without the aid of extracted rules, LLMs typically rely solely on the literal content of the document, extracting only obvious relational triples while missing more implicit ones. As shown in Figure 4(a), the LLM can infer the relationship ("Ariel Sharon", "head_of_gov", "Israel") based on the document's semantics. However, it misses the implicit relationship ("Ariel Sharon", "citizen_of", "Israel"). By providing the LLM with the rule "head_of_gov → citizen_of", our method helps the LLM extract this additional, less obvious relation. This demonstrates how our rule-based approach enables LLMs to more comprehensively complete the relation extraction task by accounting for logical patterns that might otherwise be overlooked.

In the log-based anomaly detection task, LLMs can struggle due to insufficient domain knowledge, leading to hallucination issues. In Figure 4(b), the log sequence lacks clear semantic indicators of an anomaly, making it difficult for the LLM to detect. Our method uses MCTS to extract rules from historical logs that indicate abnormal patterns. When processing a sample, the log sequence is matched with the rule base, and the corresponding rule, along with its confidence score, is provided to the LLM. This enables the LLM to combine semantic information with historical patterns and rule reliability to make accurate anomaly detections. In this case, Rule 1 triggered by "E11, E28" indicates a high probability of anomaly, allowing the LLM to correctly assess the system state.

In the decision-making task (Figure 4 (c)), the vanilla LLM only takes as input Bob's observation, therefore has a straightforward policy to walk towards the treasure directly. However, *RuAG* awares

Table 8: Searched rule examples across different tasks.

| Task | Rule | Description |
|------|------|-------------|
| Relation Extraction | head_of_gov → citizen_of | If a person holds the position of head of government, they are also a citizen of that country. |
| | head_of_gov-x → citizen_of-x | If a person holds the position of head of government in a nominal variation of a country, they are also a citizen of that nominal variation of the country. |
| Anomaly Detection | [E11 & E28] → abnormal, precision = 0.96 | If events E11 and E28 occur sequentially, it indicates a high probability of anomaly with a precision of 0.96. |
| | [E11 & E26 & E20] → abnormal, precision = 0.99 | If events E11, E26, and E20 occur sequentially, it indicates a very high probability of anomaly with a precision of 0.99. |
| Cooperative Game | [IsGreen(Alice, Left) & Move(Alice, right) & $d_x$(Alice, treasure)=0 & $d_x$(Alice, treasure) & Stand(Bob, skyblue) & Visited(Bob, skyblue) & Visited(Bob, purple) & Visited(Alice, yellow) ] → Reward = 100.0 | When Alice's center left block is green, if Alice moves right, then the team will receive a Reward = 100.0. In all these cases, Alice locates at 0 blocks down, 1 block to the left of the treasure, Bob stands on skyblue block, Bob visited skyblue block, Alice visited yellow block, Bob visited purple block. |

(a) Relation Extraction

(b) Log-based Anomaly Detection

(c) Cooperative Game

Figure 4: Case studies on relation extraction, log-based anomaly detection, and cooperative game.

Bob the domain-specific knowledge: to stand on the skyblue block is a significant step for your team's success. Therefore, in *RuAG*, Bob chooses to walk to the skyblue block first. This cooperative game highlights the significance of domain-specific knowledge in decision-making tasks, and demonstrates the effectiveness of our *RuAG* to integrate domain-specific knowledge by logic rules.

## 5 CONCLUSION

In this paper, we introduce a novel framework *RuAG* that automatically distills large volumes of offline data into understandable first-order logic rules, which are then injected into LLMs to enhance their generation capabilities. By leveraging LLMs' commonsense, we first automatically formulate the searching process through defining the target predicate and body predicates. Then, we apply Monte Carlo Tree Search (MCTS) to efficiently address the combinatorial search space. As a consequence, our method discovers logic rules that can be seamlessly integrated into LLM prompts for downstream task reasoning. Empirical evaluations across a variety of tasks, including NLP, time-series, decision-making, and industrial applications, demonstrate the effectiveness of our approach in improving LLM performance over diverse domains.

ETHICS STATEMENT

In this paper, we strictly obey the principles outlined in the ICLR Code of Ethics, including careful consideration of potential ethical concerns, including the impact on human subjects, data privacy, and fairness in algorithmic decisions. Specifically, the three public datasets do not have potential risk. As for the private industrial dataset, we promise that any data used in this study was released in compliance with legal and ethical standards, and proper security measures were implemented to safeguard personal information.

REPRODUCIBILITY STATEMENT

We provide all the details of our method in the paper and appendix, including evaluation prompts, detailed experimental setup, implementation, and hyperparameters for both LLM reasoning and MCTS. The code will be available upon the paper's publication. The above ensures that others can reproduce our method.

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

## A   EXPERIMENTAL RESULTS ON PRIVATE INDUSTRAIL DATASET: UNAUTHORIZED PARTY ABUSE DETECTION

The Unauthorized Party Abuse (UPA) detection task is a binary classification problem, where the goal is to predict whether an incident is a case of UPA (IsUPA) based on a series of features. These features include both time-dependent data, such as resource acquisition velocities and user activity history, as well as static features, like resource descriptions and types of compromised subscriptions. The task is to accurately classify each event as either UPA or not, while maintaining high precision and recall to avoid misclassifying legitimate customer activities.

**Setup**   The dataset used for this task comes from a private industrial source, consisting of historical incidents of Unauthorized Party Abuse (UPA). It includes both time-dependent features, such as resource acquisition velocities and user activity history, as well as static features, like resource descriptions and types of compromised subscriptions. The dataset is imbalanced, with significantly fewer UPA cases compared to legitimate ones, and the overall data volume is large. To address this, we sampled a balanced dataset and tested the algorithm on smaller batches. For evaluation, we used common fraud detection metrics, including F1-score, Recall, Precision, and Accuracy. We compared our method against several baselines, including XGBoost, Decision Tree, and Rule Grounding. In Rule Grounding, the extracted rules were directly used for prediction to evaluate the effectiveness of rule extraction.

**Implement Details**   In our task, most features in the dataset are continuous. To adapt to the requirement of Monte Carlo Tree Search (MCTS) for discrete state mining, we used the Gini index to discretize these continuous features. Specifically, for each continuous feature, we divided it into 10 discrete states. The discretization process involved calculating the Gini index to determine the optimal split points, ensuring that each resulting interval maintains a high degree of data purity. Thus, each data sample was converted into a sequence of discrete states.

We used Monte Carlo Tree Search (MCTS) to extract rules from the training set. MCTS was initialized with a root node representing the initial state. Child nodes were created and expanded using the Upper Confidence Bound (UCB) formula. Simulations were performed to explore different paths, and optimal rules were generated for both IsUPA=1 and IsUPA=0 targets. The rollout was set to 500, and the reward was based on the precision derived from the rule. The maximum rule length was set to 5. Additionally, if a node's precision exceeded 0.85, we considered it a terminal node, as further expansion was deemed unnecessary. This allowed us to collect all reasonable rules with lengths ranging from 1 to 5.

**Main result**   Table A1 shows the results of different methods on the small batch dataset for abuse detection. We observe that the rules extracted using MCTS achieve high precision, similar to traditional machine learning methods, but also exhibit a higher recall. This is because MCTS explores a broader search space, allowing it to capture a more comprehensive set of abuse patterns. On the other hand, directly using the LLM for this task yields poor performance, with an F1 score of only 22.64%. The lack of domain-specific knowledge and the difficulty in processing purely numerical features hinder the LLM's effectiveness in this scenario.

However, our method, which provides the MCTS-extracted rules as historical guidance to the LLM, enables the LLM to make better decisions by combining the extracted rules with feature information from specific scenarios. The results indicate that our approach significantly improves the LLM's performance on this type of numerical task. With the help of rules, the LLM's F1 score increases to 96%, demonstrating the effectiveness of our method in guiding the LLM to handle such tasks better. The table shows several representative rules extracted using MCTS, along with their precision, recall, and F1-score if used directly for detection. As can be seen, just using the first rule alone yields an F1 score of 0.6623. Additionally, precision is crucial for rules in this task, as high precision means that the rule for predicting IsUPA=1 is highly reliable and unlikely to make false positive errors.

Table A1: Comparison under different methods on Fraud detection

|  | F1 | Precision | Recall |
|---|---|---|---|
| Decision tree | 83.72% | 100% | 72% |
| XGBoost | 88.89% | 100% | 80% |
| Rule grounding | 93.62% | 100% | 88% |
| Vanilla | 22.64% | 21.43% | 24% |
| Ours | 96% | 96% | 96% |

Table A2: Representative rule, precision, and description of unauthorized party abuse detection.

| Conditions | Target | Precision | Recall | F1 |
|---|---|---|---|---|
| `Feature1` ≤ 0.030 and `Feature2` is 1 and 0.003 < `Feature3` ≤ 0.547 | 1 | 0.8632 | 0.5372 | 0.6623 |
| 0.348 < `Feature4` ≤ 0.712 | 1 | 0.8229 | 0.4202 | 0.5563 |
| `Feature1` ≤ 0.030 and `Feature2` is 1 and 0.258 < `Feature4` ≤ 0.348 | 1 | 0.9630 | 0.1383 | 0.2419 |

## B MORE EXAMPLES OF SEARCHED RULES

We provide the searched rules in Table A3 (Relation Extraction), Table A4(Log-based anomaly detection), Listing 1(Cooperative game) and Table A2 (Abuse detection).

Table A3: Representative rule, precision, and description of relation extraction

| Rule | Precision | Description |
|---|---|---|
| player_of→member_of | 1.0 | If someone is a player of a certain team, then they are also a member of that team. For example, "John is a player_of TeamA" can be deduced as "John is a member_of TeamA". |
| minister_of→agent_of | 0.9928 | If someone is a minister of a certain organization or country, then they are also an agent of that organization or country. For example, "Alice is a minister_of Country_X" can be deduced as "Alice is an agent_of Country_X". |
| head_of_state-x, gpe0 → head_of_state | 0.7472 | If someone is the head of state of a nominal variation of a country, and that nominal variation corresponds to an official country name, then they are also the head of state of that country. For example, "PersonA is the head_of_state-x of German" and "German is gpe0 of Germany" can be deduced as "PersonA is the head_of_state of Germany". |
| head_of_gov, in0-x → citizen_of-x | 0.8235 | If someone is the head of government of a country, and a geographic location in that country has a nominal variation, then the head of government can be considered a citizen of the nominal variation. For example, "PersonB is the head_of_gov of Israel" and "Tel Aviv is in0-x of Israeli" can be deduced as "PersonB is citizen_of-x of Israeli". |
| head_of, agency_of → citizen_of | 0.6364 | If someone is the head of an organization, and that organization is an agency of a country, then the head of the organization can be considered a citizen of that country. For example, "PersonC is head_of Organization_Y" and "Organization_Y is agency_of Country_Z" can be deduced as "PersonC is citizen_of Country_Z". |

```
1) Summarized experiences related to **Bob stands on yellow block**
   - Conditions: Alice visited yellow block, Bob visited purple block, and Bob visited skyblue block.
   - When Bob locates at 5 blocks down and 0 block to the left of the treasure, if Bob moves down, then Bob
     will stand on yellow block.
2) Summarized experiences related to **Bob stands on purple block**
   - When Bob locates at 2 blocks down and 9 blocks to the left of the treasure, if Bob moves right, then Bob
     will stand on purple block.
   - When Bob locates at 1 block down and 8 blocks to the left of the treasure, if Bob moves down, then Bob
     will stand on purple block.
   - When Bob locates at 2 blocks down and 8 blocks to the left of the treasure, if Bob keep standing on
     current block, then Bob will stand on purple block. In all these cases, Bob visited purple block.
   - When Bob locates at 2 blocks down and 8 blocks to the left of the treasure, if Bob moves right, then Bob
     will stand on purple block. In all these cases, Bob visited purple block.
   - When Bob locates at 2 blocks down and 8 blocks to the left of the treasure, if Bob moves down, then Bob
     will stand on purple block. In all these cases, Bob visited purple block.
3) Summarized experiences related to **Alice stands on skyblue block**
   - Conditions: Alice visited yellow block, and Bob visited purple block.
   - When Alice locates at 0 block down and 5 blocks to the left of the treasure, if Alice moves left, Bob
     did not visit skyblue block, then Alice will stand on skyblue block.
4) Summarized experiences related to **Alice stands on green block**
   - Conditions: Bob stand on skyblue block, and Bob visited skyblue block, Alice visited yellow block, Bob
     visited purple block
   - When Alice locates at 1 block down and 0 block to the left of the treasure, if Alice moves up, then
     Alice will stand on green block.
   - When Alice locates at 0 block down and 1 block to the left of the treasure, if Alice moves right, then
     Alice will stand on green block.
5) Summarized experiences related to **Alice stands on yellow block**
   - Conditions: Bob visited purple block
   - When Alice locates at 6 blocks down and 0 block to the left of the treasure, if Alice's action is not up
     , Alice's action is not left, then Alice will stand on yellow block. In all these cases, Alice visited
     yellow block.
   - When Alice locates at 6 blocks down and 1 block to the left of the treasure, if Alice moves right, then
     Alice will stand on yellow block.
   - When Alice locates at 5 blocks down and 0 block to the left of the treasure, if Alice moves down, then
     Alice will stand on yellow block.
   - When Alice locates at 6 blocks down and 0 block to the left of the treasure, if Alice keep standing on
     current block, then Alice will stand on yellow block. In all these cases, Alice visited yellow block.
   - When Alice locates at 6 blocks down and 0 block to the left of the treasure, if Alice moves down, then
     Alice will stand on yellow block. In all these cases, Alice visited yellow block.
   - When Alice locates at 6 blocks down and 0 block to the left of the treasure, if Alice moves right, then
     Alice will stand on yellow block. In all these cases, Alice visited yellow block.
6) Summarized experiences related to **Bob stands on skyblue block**
   - Conditions: Alice visited yellow block, and Bob visited purple block.
   - When Bob locates at 0 block down and 5 blocks to the left of the treasure, if Bob moves left, Alice does
     not stand on skyblue block, then Bob will stand on skyblue block.
   - When Bob locates at 0 block down and 5 blocks to the left of the treasure, if Alice's action is not left
     , Bob moves left, then Bob will stand on skyblue block.
7) Summarized experiences related to **the team receive a Penalty of -10.0 reward**
   - Conditions: Bob stands on skyblue block, Bob visited skyblue block, Alice visited yellow block, Bob
     visited purple block, Bob's action is not stand.
   - When Alice's upper right block is green, Alice's action is not down, if Bob moves right, then the team
     will receive a Penalty of -10.0 reward. In all these cases, Alice locates at 1 block down and 1 block to
     the left of the treasure.
   - When Alice locates at 1 block down and 1 block to the left of the treasure, if Alice's action is not
     down, Bob moves right, then the team will receive a Penalty of -10.0 reward.
8) Summarized experiences related to **the team receive a Reward = 100.0 (Game Win) **
   - Conditions: Bob stands on skyblue block, Bob visited skyblue block, Alice visited yellow block, Bob
     visited purple block
   - When Alice's center right block is green, if Alice moves right, then the team will receive a Reward =
     100.0. In all these cases, Alice locates at 0 block down and 1 block to the left of the treasure.
   - When Alice locates at 0 block down and 1 block to the left of the treasure, if Alice moves right, then
     the team will receive a Reward = 100.0.
```

Listing 1: Searched rules in Alice&Bob Scenario

Table A4: Representative rule of Log-based anomaly detection

| Rule | Precision | Description |
|---|---|---|
| E7,E15 → abnormal | 1.0 | If events E11 and E28 occur sequentially, it indicates a high probability of anomaly with a confidence of 100%. |
| E11,E28 → abnormal | 0.9553 | If events E11 and E28 occur sequentially, it indicates a high probability of anomaly with a confidence of 95.53% |
| E11,E26,E20 → abnormal | 0.99 | If events E11 and E28 occur sequentially, it indicates a high probability of anomaly with a confidence of 99% |

## C  IMPLEMENTATION DETAILS

We provide detailed implementation for the three public tasks and the hyperparamter in Table A5.

### C.1  RELATION EXTRACTION

We employed Monte Carlo Tree Search (MCTS) for relation extraction across all relation triplets in the training set. The rules corresponding to terminal nodes were saved, and only those with a precision greater than 0.5 were retained, resulting in a final set of 20 rules. During decision-making, the LLMs select the most relevant rule based on similarity for each input. We experimented with both GPT-3.5 (gpt-35-turbo-16k-20230613) and GPT-4 (gpt-4-20230613). For more hyper-parameters, please refer to Table A5.

### C.2  LOG-BASED ANOMALY DETECTION

For our experiments, we sampled 20,000 blocks of log sequences from the large HDFS dataset, which contained nearly 486,060 log entries. We split the dataset in a time-ordered fashion into training, validation, and test sets with a ratio of 8:1:1. Both the sequential and semantic information of log events were used for anomaly detection. In this task, we defined rules such that if a subset of events (e.g., $E_m, E_n, E_l \rightarrow abnormal$) appears in order in a sequence, it indicates an abnormal log sequence. For example, the rule $E_m, E_n, E_l \rightarrow abnormal$ indicates that if $E_m, E_n, E_l$ appear in order within a sequence, the sequence is identified as having abnormal characteristics. We employed MCTS to search for rules in the log event sequences of the training set, with the rule's accuracy serving as the reward. During anomaly detection, both event sequence and semantic information are input into the LLM, and matching rules are retrieved from the rule library. If no matching rule is found, the LLM is notified that the log sequence does not reflect any known abnormal patterns from historical data.

### C.3  ALICE&BOB SCENARIO

We choose the cooperative puzzle-solving game *Alice&Bob* (shown in Figure A6), as it is both challenging in requiring planning and collaboration, where two agents, Alice and Bob, navigate a 13x9 grid to find a treasure (Chen et al., 2024b) and the optimal path for them are both against to the intuition. Each agent starts at different positions and can move up, down, left, right, or keep stand, constrained by walls and map boundaries. Keys open corresponding doors, and a lever removes walls, unlocking new areas. The agents only receive rewards upon reaching the treasure (+100), with penalties for hitting walls (-0.1 for general walls, -10 for removable ones). Each agent has limited visibility (a 3x3 area), and they must cooperate, using their abilities to overcome obstacles. Episodes last up to 50 steps.

The observation of the agents includes their surrounding 8 blocks, their relative distance to the treasure, their teammate's relative distance to the treasure, as well as the special blocks they visited. The candidate body predicates, including the agents' observations and their actions. We search the logic rules from different aspects following the LLMs' suggestion: 1) team reward = -10; 2) Alice or Bob stand on yellow, purple, skyblue blocks; 3) Game Win.

You are a **relation extraction assistant**, and your task is to extract specific relationships between given entities from a document. The format for a relationship triple should be (entity1, relation, entity2), for example, ('University of Cologne', 'based_in', 'Germany'). I will supply you with a document, 20 relationships with their descriptions, and the entities whose relationships need to be uncovered. Your mission is to sift through the document and extract all potential relationships between the given entities, based on the content of the document.

#### Task ####
You need to extract the relationships mentioned below. Here are the descriptions and explanations of these relationships:
{{relationships}}

To improve Recall and precision in relationship extraction, we apply a set of logic rules to deduce additional relationships based on the ones already identified. You can follow these logic rules to find more relationships between entities:
{{rules}}

Remember, the goal is to use these rules to fill in missing information and enhance the accuracy of relationship extraction. Apply these rules systematically to every piece of information you process. Please use the logical rules to derive more comprehensive relation triples as far as possible. At the same time, the relation triples inferred using Logic rule should be identified and distinguished from the original triples.

1. I have given you the following relationship triples. Based on these and the provided logical rules, derive additional relationship triples.
2. Explain your derivation process and the logical rules you applied.

####Input####
## Entities: {{Entities}}
## Document: {{Document}}

Now, based on the relationships, Document, and specified Entities I provided, extract the triples from the Document that include these Entities and relationships, and briefly state the reason for each extraction. Let's think step by step.

#### Output ####
## result:
//Please return the relationship triples in the following JSON format, and after each relation you can attach a reason:
{ ('entity1', 'relation1', 'entity2')//Reason: After each relation triple you can attach a reason.
. . .
('entity1', 'relation2', 'entity3')//Reason:
}
To summarize, your task is to extract relation triples from the given document and follow logical rules to get a more comprehensive relation triple, focusing only on the entities and relationships mentioned. Please ensure that you do not extract any duplicate triples, and you should only extract triples that involve the entities and relationships provided by me. Output the triples in the strict format (entity1, relation, entity2), such as (University of Cologne, based_in0, Germany).

Figure A1: Instruction prompt template for generating relation extraction triples.

I'm fetching logical rules between relationships. I need you to help me complete the relationship pre-processing of rule generation.
The logical rule is of the form:
- relation1 → relation3, which means that if relation1 exists for entity A and entity B , then relation3 exists for entity A and entity B.
- relation1, relation2 → relation3, which means that if relation1 exists for entity A and entity B and relation2 exists for entity B and entity C, then relation3 exists for entity A and entity C.
Given the following 20 relations and their interpretation, I need you to tell me which relations are candidates for which it is completely impossible to derive relation 3 when each is a target relation. relationships
In fact, I need you to eliminate some options to help me reduce the amount of calculation when I extract rules. It's important to make the decision as carefully as possible so that you don't miss a rule that will lead to the target relationship.
You need to find as much as possible and not miss any potential relationships. Return a json format with a reason after each item, like this:
{
relation3: [relation2, relation1], //reason: Explain why relation1, relation2 does not derive relation3
...
}

Figure A2: Instruction prompt template 1 for removing impossible body predicates in relation extraction.

I need your assistance in completing the preprocessing for generating logical rules between relationships.
The logical rules follow this format (where the predicates before the arrow are considered Body predicates and the ones after the arrow are Head predicates):

- relation1 → relation3: This means if relation1 exists between entity A and entity B, then relation3 also exists between entity A and entity B. - relation1, relation2 → relation3: This means if relation1 exists between entity A and entity B, and relation2 exists between entity B and entity C, then relation3 exists between entity A and entity C.

Given the following twenty relations and their descriptions, I need you to identify which relations are suitable for being Body predicates. Please remove the ones that are not appropriate for Body predicates. {relationships}

Please return the results as a dictionary where the key represents the relations suitable as Body predicates, and the value explains why.

Figure A3: Instruction prompt template2 for removing impossible body predicates in relation extraction.

You are a helpful assistant. Your task is to build body predicate and the head predicate for searching logic rules of a game through the Monte Carlo Tree Search.

**Game Description:**
Two player, Alice and Bob are collaborating to obtain a treasure in the grid world. At each timestep, the agent can observe their own parritial observation of the grid world, consisting of the following information:
1) the color of the blocks surrounding the agent
2) the color of the block where the agent is located
3) the relative position of the agent's teammate to the treasure
4) the relative position of the agent to the treasure

Each agent can choose to move up, move down, move right, move left and no action. If move towards an unmovable block, the agent gets a penalty of reward = -0.1. The two agents share a team reward whose values fails in [-0.2, -0.1, 0, -10, -10.1, 99.9, 100].
There are blocks with five colors: movable white blocks, unmovable blocks, movable yellow blocks, movable purple blocks, movable skyblue blocks and the green blocks representing the treasure. Yellow, purple, skyblue blocks are with different functions but we do not know what will happen if any of the agent stand on any blocks. At any timestep, the agents can not stand on a same block.

**Definitions of Logic Rule:**
The logic rules are defined as:
[body predicates: (feature 1 satisfies condition 1) & (feature 2 satisfies condition 2) ...] → [head predicate: a special game state].
As an example, (relative x-position of agent equals value 1) & (relative y-position of agent equals value 2) → Alice obtains green block.

**Your Task: Define Head Predicates:**
To help the agents know the environment better, please suggest all events that may significant for the game win. Below are some examples,
- if alice obtain treasure
- if bob obtain the green block
- the agent get a reward of -10.

Please think step by step to finish your task.

Figure A4: Instruction prompt template for suggesting new target predicates in cooperative game.

| Phase | Parameter | Relationship Extraction | Anomaly Detection | Abuse Detection | Alice&Bob |
|---|---|---|---|---|---|
| Rule Generation | Total rollouts | 500 | 500 | 500 | 500 |
| | Reward metric | Precision | F1-score | F1-score | Precision + Recall |
| | Maximum body predicates | 2 | 5 | 5 | 10 |
| | Terminal condition | Precision > 0.9 | Precision > 0.9 | Precision > 0.85 | Precision = 1 |
| LLM Reasoning | Maximum tokens | 1000 | 1000 | 1000 | 1000 |
| | Temperature | 0 | 0 | 0 | 0 |
| | Top-p | 1 | 1 | 1 | 1 |
| | Frequency penalty | 0 | 0 | 0 | 0 |
| | Presence penalty | 0 | 0 | 0 | 0 |

Table A5: Summary of MCTS Parameters and LLM Configuration Across Tasks

You will see a complete log event sequence from a Block in the HDFS file system. I will also provide you with the content of each log event in this sequence. Based on the current log sequence, you need to predict whether the system is in a [Normal] or [Abnormal] state, along with a written description of your reasoning.
## Input
 The log sequence window requiring anomaly detection is:
{logs}
 The content of each log event in this sequence is as follows:
{event_content}
## The Guidelines for anomaly detection is :
{{guidelines}}
The provided guidelines are very reliable. You need to trust the guidelines I provide to you first, unless there is more obvious and direct evidence to the contrary. If there are obvious unusual messages in your logs like "error," "failure," "exception," and so on, you can judge for yourself
The provided guidelines are very reliable. You need to trust the guidelines I provide to you first, unless there is more obvious and direct evidence to the contrary. If there are obvious unusual messages in your logs like "error," "failure," "exception," and so on, you can judge for yourself.

## And you should answer:
'System State:[Normal]' or 'System State:[Abnormal]'

You should first provide a brief explanation of your evaluation, and then always end your response with either 'System State:[Normal]' or 'System State:[Abnormal]' verbatim.

Figure A5: Instruction prompt template for Log-based anomaly detection

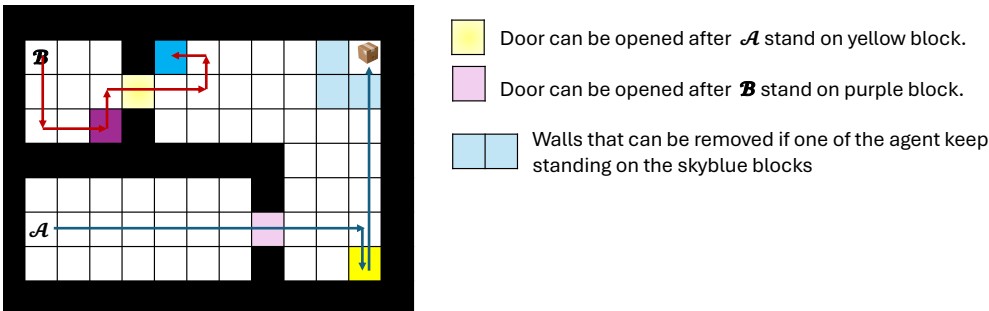

Figure A6: Illustration of Alice& Bob.

You are {agent's name}, currently collaborating with your teammate, {teammate's name}, in a grid world to obtain the treasure (green block). The final goal of your team is to secure the treasure through cooperation. Your team's performance will be evaluated based on the total rewards you collect during the game and the number of steps taken to find the treasure. Due to the impossible communication with your teammate, please monitor the state of your teammate and adjust your plan in time.
## Game Win: You or {teammate's name} reaches the treasure. Please actively collaborate with your teammate to achieve the goal.
## Candidate actions: 'up': move to stand on your **upper center** block if not black; 'down': move to stand on your **lower center** block if not blackk; 'left': move to stand on your **center left** block if not blackk; 'right': move to stand on your **center right** block if not blackk; 'stand': keep standing on the current block. Be careful to stand on the same block for a long time.
## Explanation about your surrounding blocks: - Center left, center right, upper center, lower center blocks: you can only move to any of them as long as they are non-black blocks; otherwise, you will receive a penalty and stay on original block. - Upper left, Upper right, Lower left, lower right: You need move twice to reach those blocks. So if you want to move to those blocks, please be careful to plan the path and make sure all the blocks in the path are movable. As an example: if you want to move up then right, please make sure both center right and upper center blocks are reachable.
## Some examples to avoid obstacles: - If you want to move to the lower right block and your center right block is black, you can move down first then right if your lower center blobk is white. - If moving right would bring you closer to your destination but the 'center right block' is unmovable and 'lower center block' is movable, try moving down first, then moving left twice and finally up if applicable. Mention this in your plan if you want to do so.
{Searched Logic Rules}
Please response with your thoughts, plan, and chosen action in the following format: // Describe your initial thoughts, like analysising the key steps towards game win, identifying your subgoals, comparing your candidate actions, analysising the progress of your teammate, assessing your previous plan and making future plan. "Thoughts": "Let's think step by step! [your analysis here]",
// Make your future plan after you take action at this timestep. The plan will be the reference of your future decision making. // Do not include the current chosen action in the plan. "Plan": "[fill your future plan here]",
// Your action, make sure to choose from 'up', 'down', 'left', 'right', 'stand'. "Chosen Action": "[fill your final action choice here]"
## Your last action: {previous_action}
## Your plan at last timestep: {previous_plan}
Please reaccess your situation and make decisions based on your current observations and the previous plan. If necessary, you can choose to act without considering your plan.
## Your current observation: {current_observation}

Figure A7: Instruction prompt template for generating Alice's action in Alice&Bob.

