# OpenReview forum: "RuAG: Learned-rule-augmented Generation for Large Language Models"
_ICLR.cc/2025/Conference — ICLR 2025 Poster_

### Official Review · Reviewer_31bM · 2024-11-04

**Soundness:** 3
**Presentation:** 3
**Contribution:** 3
**Rating:** 8
**Confidence:** 3

**Summary:**

This work introduces the RuAG framework that automatically distills large volumes of offline data into logical rules , which are then included in LLM prompts to enhance their reasoning capabilities.

**Strengths:**

1. RuAG offers a scalable solution for integrating extensive domain knowledge into LLMs that improves upon RAG or SFT.
2. Model performance is tested on a wide array of tasks and show improvement on strong baselines
3. RuAG is more computationally efficient than other methods that summarize external dataset as knowledge storage, as the calls to API models only happen once during logic rule constructions.

**Weaknesses:**

1. Ablation studies in Table 5 could include RAG or SFT on open-source LLMs, as the current baselines only include COT which does not include external knowledge.
2. How do LLMs suggest new rules to explore and detect impossible body predicates? These parts seem unclear to me.

**Questions:**

1. L190-195 probably has some copy-pasting errors?

---

> ### Author Response · Authors · 2024-11-21
>
> # Response to comments from Reviewer 31bM
>
> We sincerely appreciate your recognition of our work. Below, we address your concerns point by point.
>
> ## Response to Weakness 1:
> >Ablation studies in Table 5 could include RAG or SFT on open-source LLMs, as the current baselines only include COT which does not include external knowledge.
>
> Thanks for your suggestion. We add experimental result of RAG and ICL resulton relation extraction  and log-based anomaly detection as following:
>
>
> | **Method** | **Relation Extraction (F1/Precision/Recall)** | **Log-based Anomaly Detection (F1/Precision/Recall)** |
> |------------|----------------------------------------------|------------------------------------------------------|
> | **Vanilla** | 46.94% / 69.61% / 35.41%                    | 60.10% / 47.05% / 83.16%                            |
> | **ICL**     | 50.26% / 74.09% / 38.02%                    | 69.77% / 78.95% / 62.50%                            |
> | **RAG**     | 52.30% / 78.64% / 39.17%                    | 84.32% / 98.97% / 73.46%                            |
> | **Ours**    | 60.42% / 69.44% / 53.48%                    | 92.59% / 100% / 86.21%                              |
>
>
> Due to limited computational resources, we are not able to provide SFT on open-source LLMs. Instead, we provide supervised learning results, like those DL-based method in relation extraction (i.e., CNN, BiLSTM, Bert, Context-aware in Table 2), anomaly detection (i.e., DeepLog, LogRobust in Table 3); and behavior cloning in cooperative games (Table 4).
> According to the results, our method outperforms all the methods with external knowledge, no matter through in-context information or supervised training.
>
> ## Response to Weakness 2:
> > How do LLMs suggest new rules to explore and detect impossible body predicates? These parts seem unclear to me.
>
> We sincerely appreciate this valuable concern! LLMs assist in exploring new rules and identifying impossible body predicates through task-specific guidance. We explain this separately as follows:
>
> - **Eliminating Body Predicates:** LLMs analysis the task description, logical rule definitions, and candidate predicates with their descriptions to eliminate irrelevant predicates. For example, in relation extraction, certain relations like appears_in (denoting a player's participation in an event) are filtered out by LLMs based on their semantic information, as they are irrelevant to other relations.
> - **Suggesting New Target Predicates:** LLMs are prompted to propose new target predicates according to logical rule definitions, task descriptions, and the data schema. For example, in a cooperative game, the initial task-relevant predicate might be GameWin, representing whether the agents win the game. However, after analyzing the game description and the agents' observation and action spaces, LLMs may suggest exploring logical rules involving agents standing on blocks of different colors, as these could play a significant role in understanding the regulation of the game and achieving a win.
>
> In the revised version, we modify Line 208 - 218 for better understanding and provide the relevant prompts in Figure A3 and A4 in Appendix.
>
> ## Response to Question 1:
> > L190-195 probably has some copy-pasting errors?
>
> Thanks for appointing this. We will revise it in the future version.

---

> ### Author Response · Authors · 2024-11-25
>
> Dear Reviewer 31bM,
>
> Thank you for your positive support to our work. We greatly appreciate the opportunity to refine our work through your comments.
>
> We sincerely hope that our responses have effectively addressed all your questions and concerns. If there are any additional questions, we are more than happy to provide further input.
>
> Best regards,
> The Authors

---

> > ### Comment · Reviewer_31bM · 2024-11-25
> > **Thank you for your response**
> >
> > Dear Authors,
> >
> > Thank you for your thorough response. I have no further questions.

---

### Official Review · Reviewer_ME5G · 2024-11-04

**Soundness:** 3
**Presentation:** 2
**Contribution:** 2
**Rating:** 6
**Confidence:** 3

**Summary:**

This paper proposes to use LLM to create abstract rules that can be provided in context for better decision making when using LLM. The method contains a few steps where first the LM is used to define the features and labels for which we would like to forumlate rules over. Then MCTS is used to find the rules that explain the data best, and last the rules are provided in-context for prediction. The idea of using LLM to create features to search over given data is very nice. The method is evaluated on tasks of relation extraction, anomaly detection and a multi-agent game. Using GPT-4 is much better than weaker models (unsurprisingly) and using the rules is better than some prior work termed HtT (which needs more explanation). In the game experiment there is advantage over ICL and RAG.

**Strengths:**

* The idea of using large language models for "feature extraction" is very interesting. It is related to https://arxiv.org/abs/2409.08466 (which I don't expect to be in the paper since it's very recent)

* The empirical results show that for some use cases abstract rules can be leveraged to improve performance.

**Weaknesses:**

* The paper needs better **scoping** -- when is this method likely to be useful and when not? For example, the point of machine learning is that some things are hard to express by rules -- for example, what makes a face a face or a cat a cat? We learn machine learning models for cases where rules are hard to formulate. Is this method restricted to things that can be defined by rules or not? I think it's important to discuss this. Second - the method assumes an input of N features with some feature description. Often nowadays we work with more raw data like a sequence of words etc. Is this restricted to such cases? The tasks chosen are diverse but rare and it is not clear how general the method is and when we should expect it to work? Overall the generality of the method remains unclear and needs further discussion.

* Line 188: it is not clear if the LM looks at the actual data. The authors talk about using the LLM to look at the data and find patterns, but it seems maybe the LLM is only used to look at the schema, or the featuer descriptions to define what are the body and target predicates and it does not use the actual values of these features in the data at all? Choosing the rules is done with MCTS where it is not clear if the LM is used - it seems like rules are applied on the data to see if they work well. So if the LLM is not used to find patterns in the data this might be a bit limited.

* The paper has many clarity issues:
** Clarity - I don't understand figure 3 well enough - it seems important but is a mix of using text and emoji without proper explanation I can kind of squint at it and guess but it seems really difficult to understand what are the details of each step.

** Clarity: line 190: “initial the features as the body predicates” - what are the features exactly? can you give some examples and intuition at this point already?

** The paper talks about "impossible body predicates". What are those? Why are they impossible? What is exactly the input provided to the LLM to perform this task (please don't send me to the appendix in author response). Similarly "suggesting new target predicates" how? What is the task given to the LLM to do that? All those seem like crucial aspects that are not explained. I can imagine the LM doing an OK job in these things with some prompt but they don't seem necessarily like something that is well defined that even humans can do with reasonable agreement.

** HtT -- this seems like a key baseline that is not explained properly.

** Line 285: During the rule extraction process, we leveraged the LLM to filter out 15% of the relationships that were unlikely to serve as valid predicates. Unclear;

** Experimental details: There is very little detail in the paper on what are the featuers/labels/target and body predicates in each of the experiments, this makes it hard to understand the task. There are a few examples in a table but this is insufficient for understanding.

* Experimental evaluation: a lot of the comparisons in the experiment are between models that have very different abilities. Comparing GPT-4 to some CNN or BiLSTM or BERT is not a fair comparison at all. Baselines should be between the same model that uses different methods for providing in-context information and this is missing.

* Experimental evaluation: The main claim in the abstract and introduction is that using rules in context is a better alternative to in-context learning and RAG - but I don't see any RAG or ICL baselines in section 4.1 or section 4.2 so how can this claim be made? Did the author try ICL and RAG in 4.1 and 4.2? or only in 4.3? It would also be good to have much more detail on what the ICL and RAG baselines look like in Section 4.3.

* Minor: lines 189 and onwards have the same few sentences twice.

**Questions:**

** Table 4 -- where do confidences come from?

---

> ### Author Response · Authors · 2024-11-21
>
> # Response to comments from Reviewer ME5G
> Thank you for acknowledging our idea and its robust empirical support. The [referenced paper](https://arxiv.org/abs/2409.08466) reinforces our argument for using interpretable predicates. We build upon this by exploiting logical rules—relationships among these predicates—to enhance the generative capabilities of LLMs. This integration is grounded in the logic that:
> 1) Logical rules represent a fundamental form of external knowledge crucial for human reasoning and are readily translatable into the natural language constructs that LLMs can understand.
> 2) As the demand for explainable AI grows, like methods centered on causality and concept bottleneck models, our approach provides a significant advancement in using interpretable data to improve LLM generation.
>
> ## Response to Weakness 1 (Scoping):
> We appreciate your suggestion to clarify the scope, which has greatly enhanced our paper. We've provided a point-by-point response to your concerns and incorporated them into the revised version. We're happy to discuss further if needed.
>
> **Response to Weakness 1.1:**
> > The paper needs better scoping -- when is this method likely to be useful and when not? For example, the point of machine learning is that some things are hard to express by rules -- for example, what makes a face a face or a cat a cat? We learn machine learning models for cases where rules are hard to formulate. Is this method restricted to things that can be defined by rules or not? I think it's important to discuss this.
>
>
> **Scope/Generality:** our work is flexible to address all the tasks where are underlying logical or structured relationships within the data, which can be distilled into explicit rules. This is widely-appeared in real world[1][2][3], which ensures the generality of our method.
>
> *Therefore, our method is versatile and not limited to things that can solely be defined by rules.*
>
> We add the above discussion in the revised version.
>
> [1] Teru, Komal, Etienne Denis, and Will Hamilton. "Inductive relation prediction by subgraph reasoning." ICML, 2020.
>
> [2] Siyuan Wang, Zhongyu Wei, Yejin Choi, and Xiang Ren. Can LLMs Reason with Rules? Logic Scaffolding for Stress-Testing and Improving LLMs. ACL, 2024.
>
> [3] Morishita, T., Morio, G., Yamaguchi, A., & Sogawa, Y. Enhancing Reasoning Capabilities of LLMs via Principled Synthetic Logic Corpus, NeurIPS 2024.
>
> **Response to Weakness 1.2:**
> > the method assumes an input of N features with some feature description. Often nowadays we work with more raw data like a sequence of words etc. Is this restricted to such cases?
>
> Thank you for your question. Our method can handle raw data, such as sequences of words. In our method, logic rules serve as a compact knowledge source for large language models, enhancing their capacity to manage such data in tasks like relation extraction and anomaly detection. This approach is akin to Retrieval-Augmented Generation (RAG); however, it utilizes compact logic rules instead of extensive knowledge bases, offering greater efficiency and significantly reducing computational overhead during both retrieval and generation phases.
>
> **Response to Weakness 1.3:**
> > The tasks chosen are diverse but rare and it is not clear how general the method is and when we should expect it to work? Overall the generality of the method remains unclear and needs further discussion.
>
> Thank you for raising this important concern. The selection of these tasks was indeed intentional to showcase the broad applicability of our methods across various domains. Far from being rare, these tasks represent common challenges in the real world. For instance, relation extraction is a fundamental problem in NLP, as demonstrated by studies [1][2]. Similarly, log-based anomaly detection is crucial in time-series analysis [3][4], and cooperative games are widely used to study decision-making processes [5][6]. Additionally, the task of unauthorized abuse detection is a critical concern in industrial settings. Together, these examples illustrate the versatility and general applicability of our approach.
>
> [1] Kunxun Qi, Jianfeng Du, and Hai Wan. 2024. End-to-end Learning of Logical Rules for Enhancing Document-level Relation Extraction. ACL 2024.
>
> [2] Teru, Komal, Etienne Denis, and Will Hamilton. "Inductive relation prediction by subgraph reasoning." ICML, 2020.
>
> [3] Gruver, N., Finzi, M., Qiu, S., & Wilson, A. G. Large language models are zero-shot time series forecasters. NeurIPS 2024.
>
> [4] Gong, Y., Luo, H., Liu, A. H., Karlinsky, L., & Glass, J. R. Listen, Think, and Understand. ICLR 2024.
>
> [5] Piatti, G., Jin, Z., Kleiman-Weiner, M., Schölkopf, B., Sachan, M., & Mihalcea, R. Cooperate or collapse: Emergence of sustainable cooperation in a society of llm agents. NeurIPS 2024..
>
> [6]Sun, C., Huang, S., & Pompili, D. (2024). LLM-based Multi-Agent Reinforcement Learning: Current and Future Directions. arXiv preprint arXiv:2405.11106.

---

> ### Author Response · Authors · 2024-11-21
>
> ## Response to Weakness 2 (LLM's involvement in RuAG):
>
> Thank you for your inquiry regarding the use of actual data and LLMs.  We address this concern point by point and welcome further discussion.
>
> **Response to Weakness 2.1:**
> > Line 188: it is not clear if the LM looks at the actual data. The authors talk about using the LLM to look at the data and find patterns, but it seems maybe the LLM is only used to look at the schema, or the featuer descriptions to define what are the body and target predicates and it does not use the actual values of these features in the data at all?
>
> To clarify, LLMs indeed analyze both the schema and the value ranges of data., which are derived from **actual data**. This process is akin to traditional feature engineering, which typically relies on human expertise to interpret and define relevant features. By harnessing the extensive commonsense knowledge that LLMs acquire during pretraining, we can significantly reduce human involvement, enhancing both the efficiency and scalability of our method.
>
> **Response to Weakness 2.2:**
>
> > Choosing the rules is done with MCTS where it is not clear if the LM is used - it seems like rules are applied on the data to see if they work well. So if the LLM is not used to find patterns in the data this might be a bit limited.
>
> To ensure clarity and address your concerns, I would like to highlight the specific roles that LLMs play in our approach, particularly in steps 2 and 4, where LLMs are instrumental in selecting the appropriate rules.
>
> 1) **LLM-aided Predicate Definition:** LLMs are used to assist in defining predicates, including suggesting new target predicates and eliminating impossible ones.
> 2) **Rule Search via MCTS (without LLMs):** MCTS is employed to search for logic rules based on the defined candidate body and head predicates. The search in MCTS is guided by designed rewards that consider the precision and recall of the rules. LLMs are not used in this phase to avoid excessive costs.
> 3) **Post-processing of Learned Rules (without LLMs):** This step involves removing duplicates and translating rules into natural language, without involving LLMs.
> 4) **Learned-Rule-Augmented Generation:**
>    - The learned rules are **explicitly chosen** to be inserted into LLMs: in cooperative games, all learned rules are directly inserted; for relation extraction and anomaly detection, rules are retrieved based on similarity, inspired by RAG.
>    - During LLMs' generation, LLMs perform **implicit rule selection:** As multiple rules are inserted, LLMs evaluate the reliability of these rules and may refine their selection, ensuring that only the most pertinent and trustworthy rules are applied.
>
> **Allowing LLMs to examine all the data is prohibitively costly and may undermine their overall understanding.** For example, approaches like HtT and PLLB in cooperative games attempt to process all data to extract rules. However, these methods encounter significant challenges with long-text comprehension, which hampers their ability to distinguish between data samples and efficiently summarize rules. Our experimental results demonstrate that MCTS offers a more effective and practical solution for extracting knowledge, addressing these limitations with greater success.
>
>
> ## Response to Weakness 3 (clarity issues):
>
> Thanks for pointing these out. We response to the clarity issues point by point as following.
>
> **Response to Weakness 3.1:**
> > I don't understand figure 3 well enough - it seems important but is a mix of using text and emoji without proper explanation I can kind of squint at it and guess but it seems really difficult to understand what are the details of each step.
>
> Thank you for bringing this to our attention. Given the constraints of limited page space, we initially chose to use emojis in the figure, recognizing that this might compromise clarity in certain illustrations. To address this concern, we have implemented the following modifications in the attached version:
> - Simplified the symbols used for players (`A` for Alice, `B` for Bob), treasure (`box`), and blocks.
> - Provided clear explanations for the symbols and predicates in natural language:
>   - Visit(A, `yellow block`): Indicates whether Alice has visited the yellow block.
>   - Stand(A, `yellow block`): Indicates whether Alice is currently standing on the yellow block.
>   - DisX(A, `box`) = -6: Represents the horizontal distance between Alice and the diamond, measured as -6.
>
> We greatly appreciate any further suggestions you may have for optimizing the figures!

---

> > ### Author Response · Authors · 2024-11-21
> >
> > **Response to Weakness 3.3:**
> > > The paper talks about "impossible body predicates". What are those? Why are they impossible? What is exactly the input provided to the LLM to perform this task (please don't send me to the appendix in author response). Similar, "suggesting new target predicates" how? What is the task given to the LLM to do that? All those seem like crucial aspects that are not explained. I can imagine the LM doing an OK job in these things with some prompt but they don't seem necessarily like something that is well defined that even humans can do with reasonable agreement.
> >
> > We would like to address your concerns point-by-point:
> >
> > **How do LLMs assist in defining predicates?**
> > - Eliminating Body Predicates: LLMs achieve this by analyzing the task description, logical rule definitions, and candidate predicates with their descriptions. For example, in relation extraction, certain relations like `appears_in` (denoting a player's participation in an event) are filtered out by LLMs based on their semantic information, as they are irrelevant to other relations.
> > - Suggesting New Target Predicates: LLMs are guided by logical rule definitions, task descriptions, and the data schema to propose new target predicates. For example, in a cooperative game, the initial task-relevant predicate might be `GameWin`, representing whether the agents win the game. However, after analyzing the game description and the agents' observation and action spaces, LLMs may suggest exploring logical rules involving agents standing on blocks of different colors, as these could play a significant role in achieving a win.
> >
> > **Can LLMs work in defining predicates?** For the body predicates elimination, we provide the initial predicates so that they just remove some of them; as for the head predicates, we found that LLMs can generate some predicates that easy to be extract from the initial predicates as well.
> >
> > We revise the Line 208 - 218 for clarity and provide detailed prompts in Figure A3, A4 in Appendix.
> >
> > **Response to Weakness 3.4:**
> > > HtT -- this seems like a key baseline that is not explained properly.
> >
> > HtT shares similar motivation that use constructed rule library to enhance LLMs' generation. They builds a rule library by having an LLM generate and verify rules over training examples. However, our method is more computationally efficient in learning rules by structuring them in a systematic manner. This enables MCTS to learn rules effectively while significantly reducing the reliance on extensive LLM calls.
> >
> > We incooperate this in the revised version (Line 264 - 286).
> >
> > **Response to Weakness 3.5:**
> > > Line 285: During the rule extraction process, we leveraged the LLM to filter out 15% of the relationships that were unlikely to serve as valid predicates. Unclear;
> >
> > To reduce searching cost in MCTS, LLMs are prompted to eliminate impossible body predicates (as said in **Response to Weakness 3.3**) according to the relation description and rule description and we found there are likely 15\% of the total body predicates candidates are eliminated in this task, including `vs`，`appears_in` and `player_of`.
> >
> > We revise this sentence for better understanding in the attached version of our paper (Line 299 - 301).
> >
> >
> > **Response to Weakness 3.6:**
> > > Experimental details: There is very little detail in the paper on what are the featuers/labels/target and body predicates in each of the experiments, this makes it hard to understand the task. There are a few examples in a table but this is insufficient for understanding.
> >
> > Thanks for pointing this out. As we explained in **Response to Weakness 3.2**, the features and labels come from the dataset, and we translate them into body predicates and head predicates as follows:
> >
> > - Relation Extraction: Features and labels refer to the relationships between entities, such as `in0(A, C)` (A is located in country C). We define the target and body predicates as follows:
> >     - Target Predicate: A chosen relation, e.g., `in0(A, C)`.
> >     - Body Predicates: Remaining relations excluding the target predicate. For example, if the target predicate is `in0(A, C)`, the body predicates include all other relations.
> > - Log-Based Anomaly Detection: Features indicate whether specific log events occurred, while labels `Anomaly` indicate whether the log sequence is abnormal.
> >     - Target Predicate:`Anomaly`
> >     - Body Predicates: Log events, such as `E5` (receiving a block) and `E7` (write operation exception).
> > - Cooperative Game: Observations and actions in the collected data are treated as features, and the labels `GameWin` indicate whether the team won.
> >     - Target Predicate: initial target predicate is task-relevant, i.e., `GameWin`
> >     - Body Predicates: Transformed from observations and actions, e.g., `IsYellow(Alice, Right)` (Alice's right block is yellow) and `Move(Bob, Right)` (Bob moves right).
> >
> > We have added Table 1 in the revised version to enable a better understanding of the features and predicates.

---

> ### Author Response · Authors · 2024-11-21
>
> **Response to Weakness 3.7:**
> > Experimental evaluation: a lot of the comparisons in the experiment are between models that have very different abilities. Comparing GPT-4 to some CNN or BiLSTM or BERT is not a fair comparison at all. Baselines should be between the same model that uses different methods for providing in-context information and this is missing.
>
> Thanks for your question. The comparison is fair since SOTA methods like CNN, BiLSTM, and BERT are trained on the provided training data, whereas LLMs are not fine-tuned on it. The experimental results support our claim that learned logic rules can enhance LLM generation, enabling it to perform competitively with or even surpass SOTA methods across various domains.
>
> We also provide baselines with different methods for providing in-context information, like  HtT in relation extraction, LogGPT in anomaly detection and Vanilla, HtT, ICL-good, ICL-contrastive, PLLB, and RAG in cooperative game.   Additionally, we have included results for Vanilla, ICL, and RAG in relation extraction (Tables 2) and anomaly detection (Tables 3) in the revised version.  **Among all the methods providing in-context learning information, our method achieves the best performance and enjoys the efficient computation.**
>
> | **Method** | **Relation Extraction (F1/Precision/Recall)** | **Log-based Anomaly Detection (F1/Precision/Recall)** |
> |------------|----------------------------------------------|------------------------------------------------------|
> | **Vanilla** | 46.94% / 69.61% / 35.41%                    | 60.10% / 47.05% / 83.16%                            |
> | **ICL**     | 50.26% / 74.09% / 38.02%                    | 69.77% / 78.95% / 62.50%                            |
> | **RAG**     | 52.30% / 78.64% / 39.17%                    | 84.32% / 98.97% / 73.46%                            |
> | **Ours**    | 60.42% / 69.44% / 53.48%                    | 92.59% / 100% / 86.21%                              |
>
> **Response to Weakness 3.8:**
> > Experimental evaluation: The main claim in the abstract and introduction is that using rules in context is a better alternative to in-context learning and RAG - but I don't see any RAG or ICL baselines in section 4.1 or section 4.2 so how can this claim be made? Did the author try ICL and RAG in 4.1 and 4.2? or only in 4.3? It would also be good to have much more detail on what the ICL and RAG baselines look like in Section 4.3.
>
> We would like to address your concern point by point.
>
> **More baselines in relation extraction and anomaly detection:** Thank you for your suggestions. We have added experimental results for Vanilla, ICL, and RAG in both relation extraction and anomaly detection, as shown in the table in **Response to Weakness 3.7**. According to the experimental results, our method outperforms Vanilla, ICL, and RAG across diverse tasks, supporting our claim that RuAG is a viable alternative to ICL and RAG.
>
> **More implementation details on baselines in cooperative game:** In the cooperative game task (Sec 4.3), we provide additional details on how different methods deliver in-context knowledge: `ICL-good` prompts LLMs with three good demonstrations; `ICL-Contrastive` provides external knowledge through two good and two bad demonstrations; and RAG retrieves timesteps with similar observations, informing LLMs of both observations and actions.
> As discussed in Lines 409–411, due to the lengthy trajectory descriptions in the game, `ICL-good` and `ICL-Contrastive` often struggle to interpret the examples effectively, frequently failing to act intuitively—such as moving directly toward the treasure. Additionally, RAG sometimes retrieves poor action samples, which can mislead the LLMs.
>
> We revise Sec. 4.3 in the attached version to include above.
>
> **Response to Weakness 3.9:**
> > Minor: lines 189 and onwards have the same few sentences twice.
>
> Thanks for appointing this. We address this in the revised version.
>
> ## Response to Question 1:
> > Table 4 -- where do confidences come from?
>
> The 'confidence' in Table 4 refers to the rule's  precision by grounding it in the training set. We assume that a rule with high precision is reliable and pass this information to LLM for more considering the uncertainty in learned rules. We replace the term 'confidence' with 'precision' in the revised version for better understanding.

---

> > ### Comment · Reviewer_ME5G · 2024-11-22
> > **Thanks for the response**
> >
> > In light of the additional baseline I will raise my score.

---

> > > ### Author Response · Authors · 2024-11-22
> > >
> > > We appreciate your positive feedback and recognition of our work!

---

### Official Review · Reviewer_fTJ7 · 2024-11-05

**Soundness:** 2
**Presentation:** 2
**Contribution:** 2
**Rating:** 5
**Confidence:** 4

**Summary:**

The paper presents a rule-augmented generation approach, where rules are learned from the training dataset using Monte Carlo Tree Search (MCTS).
It shows that leveraging rules can outperform retrieved passages or even other supervised trained models.

**Strengths:**

The paper contributes by integrating rule-based augmentation with generation models, leveraging rules learned from data.

**Weaknesses:**

There are several areas of concern:

1. Clarity in Section 3.1 (LLM-based Logic Rule Search): This section is difficult to understand. Here are some follow-up questions for clarification:

    1.1. What do the initial predicates look like across the three different datasets?

    1.2. How does the LLM eliminate impossible predicates? Could you provide prompt examples?

    1.3. How does the LLM propose new target predicates? Any prompt examples for this?

2. Performance of Rules Alone: It appears that in cases where the rules generalize well to the test set, predictions might be straightforward using only the rules. However, this may not extend to more complex or varied test cases.

3. Applicability of Rules: The rules generated in this paper may not directly translate to real-world retrieval-augmented generation (RAG) settings, which often require "external knowledge" represented in both body and target predicates. For instance, the example in Figure 2 does not reflect how real-world decisions about weather predictions are made. The rules discussed here seem applicable primarily to scenarios with deterministic target predicates (e.g., classification tasks). How might this approach be extended to real-world scenarios, as suggested in the conclusion?

**Questions:**

1. Did you use only the "train" split to construct the rules, testing them exclusively on the "test" split?
2. How did you partition the data for the cooperative game into train, validation, and test sets?
3. Given the reliance on deterministic predicates, how do you anticipate this approach adapting to real-world RAG scenarios requiring dynamic, knowledge-based decision-making?

---

> ### Author Response · Authors · 2024-11-21
>
> # Response to comments from Reviewer fTJ7
> Thank you for your constructive review to help further improve our paper. Below, we provide a point-by-point response to your concerns.
>
> ## Response to Weakness 1 (Clarity in Section 3.1):
> **Response to Weakness 1.1:**
> > 1.1. What do the initial predicates look like across the three different datasets?
>
> Thank you for question. A predicate is a function or condition that evaluates input and returns a boolean value. A logic rule can be represented as `BodyPredicate1=True & BodyPredicate2=True & ... -> TargetPredicate=True`. Initial predicates are derived by converting dataset features into binary variables. Below are more details for each task:
> - Relation extraction: This task identifies relationships among entities within a paragraph. The dataset contains 20 distinct relations, such as `in0(A, C)` (A is located in country C).
>     - **Target predicate**: The specific relation being predicted, e.g., `in0(A, C)`.
>     - **Body predicate**: Remaining relations excluding the target predicate. For instance, if the target predicate is `in0(A, C)`, body predicates are all other relations in the dataset.
> - Log-based anomaly detection: This task classifies whether a sequence of log events isabnormal. Each sequence comprises log events such as `E5` (receiving a block).
>     - **Target predicate**: `Anomaly`, which indicates whether the log sequence is abnormal.
>     - **Body predicates**: Idividual log events, like `E5` (receiving a block) and `E7` (write operation exception).
> - Cooperative Game: In this scenario, two players collaborate to locate a treasure. At each timestep, agents take actions based on their observations, such as the color of surrounding blocks.
>     - **Target predicate**: `GameWin`, indicating whether the agents successfully win the game.
>     - **Body predicates**:  Observations and actions transformed into predicates, e.g., `IsYellow(Alice, Right)` (Alice's right block is yellow), `Move(Bob, Right)` (Bob moves right).
>
> To enhance clarity, we have added Table 1 in the revised paper, to address your concerns and make the concept of initial predicates easier to understand.
>
> **Response to Weakness 1.2:**
> > 1.2. How does the LLM eliminate impossible predicates? Could you provide prompt examples?
>
> Thank you for question. Taking the relation extraction task as an example, we utilize the LLM to reduce searching cost in MCTS by filtering out relation candidates that do not pertain to other relations. This process considers the logical rule definition and detailed relation descriptions. The LLM is prompted with the following instructions:
> ``` txt!
>     I need your assistance in completing the preprocessing for generating logical rules between relationships.
>
>     The logical rules follow this format (where the predicates before the arrow are considered Body predicates and the ones after the arrow are Head predicates):
>
>     - relation1 -> relation3: This means if relation1 exists between entity A and entity B, then relation3 also exists between entity A and entity B.
>     - relation1, relation2 -> relation3: This means if relation1 exists between entity A and entity B, and relation2 exists between entity B and entity C, then relation3 exists between entity A and entity C.
>
>     Given the following twenty relations and their descriptions, I need you to identify which relations are suitable for being Body predicates. Please remove the ones that are not appropriate for Body predicates.
>     {relationships}
>
>     Please return the results as a dictionary where the key represents the relations suitable as Body predicates, and the value explains why.
> ```
>
> We include the above prompt in Figure A3 in Appendix.

---

> ### Author Response · Authors · 2024-11-21
>
> **Response to Weakness 1.3:**
> > 1.3. How does the LLM propose new target predicates? Any prompt examples for this?
>
> Thank you for your question. In our method, LLMs suggest new target predicates based on the definitions of logical rules, task descriptions, and data schemas. For example, in a cooperative game, LLMs utilize these definitions along with descriptions of the game, agents’ observation spaces, and action spaces to propose new target predicates, as demonstrated in the prompt:
> ```txt
> You are a helpful assistant. Your task is to build body predicate and the head predicate for searching logic rules of a game through the Monte Carlo Tree Search.
>
> **Game Description:**
> Two player, Alice and Bob are collaborating to obtain a treasure in the grid world. At each timestep, the agent can observe their own paritial observation of the grid world, consisting of the following informations:
> 1) the color of the blocks surrounding the agent
> 2) the color of the block where the agent is located
> 3) the relative position of the agent's teammate to the treasure
> 4) the relative position of the agent to the treasure
>
> Each agent can choose to move up, move down, move right, move left and no action. If move towards an unmovable block, the agent gets a penalty of reward = -0.1. The two agents share a team reward whose values fails in [-0.2, -0.1, 0, -10, -10.1, 99.9, 100].
>
> There are blocks with five colors: movable white blocks, unmovable blocks, movable yellow blocks, movable purple blocks, movable skyblue blocks and the green blocks representing the treasure. Yellow, purple, skyblue blocks are with different functions but we do not know what will happen if any of the agent stand on any blocks. At any timestep, the agents can not stand on a same block.
>
> **Definitions of Logic Rule:**
> The logic rules are defined as:
> [body predicates: (feature 1 satisfies condition 1) & (feature 2 satisfies condition 2) & ...] -> [head predicate: a special game state].
> As an example, (relative x-position of agent equals value 1) & (relative y-position of agent equals value 2) -> Alice obtains green block.
>
> **Your Task: Define Head Predicates:**
> To help the agents know the environment better, please suggest all events that may significant for the game win. Below are some examples,
>      - if alice obtain treasure
>      - if bob obtain the green block
>      - the agent get a reward of -10.
>
> Please think step by step to finish your task.
> ```
>
> We include the above prompt in Figure A4 in Appendix.
>
>
>
>
> ## Response to Weakness 2:
> > Performance of Rules Alone: It appears that in cases where the rules generalize well to the test set, predictions might be straightforward using only the rules. However, this may not extend to more complex or varied test cases.
>
> We agree that directly applying rules may work well for straightforward cases but is limited in addressing more complex or varied test cases. However, our proposed method is both flexible and capable of handling such scenarios for the following reasons:
>
> **Leveraging LLMs for Complex Tasks:**
> For tasks where logic rules cannot be directly applied, our method utilizes the generation capabilities of LLMs in conjunction with logic rules. This approach mimics human reasoning, where LLMs are guided by the external knowledge encapsulated in the rules to address complex cases. Furthermore, while traditional rule-based methods struggle with raw data inputs (e.g., paragraphs of text), our method enables processing such data by relying on LLMs’ generative abilities.
>
> **Addressing Out-of-Distribution and Misleading Rules:**
> Even for tasks where logic rules are directly applicable, relying solely on rules can encounter challenges with out-of-distribution samples or inaccuracies caused by misleading rules. Our method addresses these issues by prompting LLMs to integrate relevant knowledge from multiple learned rules, enhancing the robustness and accuracy of predictions or decisions. This allows the system to generalize effectively, even when faced with unforeseen or edge-case scenarios.
>
> By combining logic rules with LLMs, our method ensures greater adaptability and robustness across a wide range of tasks, including those with complexities or variations that traditional rule-based systems cannot handle effectively.

---

> ### Author Response · Authors · 2024-11-21
>
> ## Response to Weakness 3 (Applicability of Rules):
>
> Thanks for your comments, below are our point-wise response to this concern:
>
> **Response to Weakness 3.1:**
> > The rules generated in this paper may not directly translate to real-world retrieval-augmented generation (RAG) settings, which often require "external knowledge" represented in both body and target predicates. For instance, the example in Figure 2 does not reflect how real-world decisions about weather predictions are made.
>
> Thank you for raising this point. We believe our learned rules are highly applicable to real-world retrieval-augmented generation (RAG) settings for the following reasons:
>
> ***Direct Translation to RAG Settings:*** The logic rules learned by our method are inherently interpretable and can be seamlessly translated into natural language. This capability is demonstrated in lines 73–76 of the paper. Additionally, because the rules can be expressed in natural language, they can be effectively retrieved using LLM or semantic similarity measures, making them well-suited for integration into RAG pipelines.
>
> **Clarification of Figure 2's Purpose:** We would like to clarify that Figure 2 serves as a simplified illustration of the logic rules generated by our method and is not intended to represent the full pipeline. While Figure 2 shows how rules encode external knowledge in a structured format, our approach extends far beyond this, leveraging these rules to guide large language models (LLMs) in making predictions or decisions. Specifically, the rules assist LLMs by providing external knowledge in an interpretable and accessible way, enabling them to handle complex tasks that go beyond the scope of direct rule application.
> **Response to Weakness 3.2:**
>
> > The rules discussed here seem applicable primarily to scenarios with deterministic target predicates (e.g., classification tasks).
>
> Our method is flexible and capable of addressing non-deterministic and non-classification tasks for the following reasons:
>
> **Handling Non-Deterministic Predicates:** For tasks involving non-deterministic predicates, our method learns rules with the high precision and incorporates these rules, along with their associated precision scores, into the input for large language models (LLMs). By including this information, LLMs can rely on their own reasoning capabilities to evaluate the reliability of the rules and select appropriate ones for predictions. This process enables our method to handle tasks where target predicates are not strictly deterministic or exhibit variability.
>
> **Addressing Non-Classification Tasks:** In non-classification tasks, such as decision-making or strategic reasoning, logic rules act as providers of domain-specific knowledge. For example, in game-related scenarios, rules might supply hidden information, such as identifying key blocks necessary to achieve a winning strategy. By presenting this external knowledge in an interpretable form, the rules enhance the LLM’s ability to solve complex tasks by complementing its generative and reasoning capabilities.
>
> **Response to Weakness 3.3:**
> > How might this approach be extended to real-world scenarios, as suggested in the conclusion?
>
> Our method is **highly adaptable to various real-world scenarios** for several compelling reasons:
>
> - Inspired by human reasoning processes, large language models (LLMs) excel in task resolution when bolstered by external knowledge sources [1]. Logical rules, serving as a robust framework for this knowledge, have proven universally effective across diverse real-world applications [2][3][4]. Furthermore, in practical settings, researchers often employ these rules to analyze and select features during the feature engineering phase.
> - By integrating with LLMs, our method is both flexible in handling raw data (e.g., paragraphs), addressing out-of-distribution cases, filtering inaccurate rules, and combining multiple rules seamlessly.
> - Leveraging LLMs' assistance in predicate definition, which draws on human commonsense reasoning, our method becomes more versatile across diverse tasks.
>
> To showcase its practicality, we apply our method to an industrial task—Unauthorized Party Abuse (UPA) detection—demonstrating its effectiveness and real-world applicability.
>
>
> [1] Lewis, P., Perez, E., Piktus, A., Petroni, F., Karpukhin, V., Goyal, N., ... & Kiela, D. (2020). Retrieval-augmented generation for knowledge-intensive nlp tasks. NeurIPS 2020.
>
> [2] Teru, Komal, Etienne Denis, and Will Hamilton. "Inductive relation prediction by subgraph reasoning." ICML 2020.
>
> [3] Siyuan Wang, Zhongyu Wei, Yejin Choi, and Xiang Ren. Can LLMs Reason with Rules? Logic Scaffolding for Stress-Testing and Improving LLMs. ACL 2024.
>
> [4] Morishita, T., Morio, G., Yamaguchi, A., & Sogawa, Y. Enhancing Reasoning Capabilities of LLMs via Principled Synthetic Logic Corpus. NeurIPS 2024.

---

> ### Author Response · Authors · 2024-11-21
>
> ## Response to Question 1 & 2:
> > Question 1: Did you use only the "train" split to construct the rules, testing them exclusively on the "test" split?
> > Question 2: How did you partition the data for the cooperative game into train, validation, and test sets?
>
> Yes, we adhere to standard evaluation protocols across all tasks, where we derive logic rules exclusively from the training data and assess the performance of rule-augmented LLMs using the test split. Below are the details of our training and evaluation procedures:
>
> | Task                 | Training Data for Logic Rule Learning        | Evaluation Data                                                        |
> |------------------|-------------------------------------------------|-----------------------------------------------------|
> | Relation Extraction  | Training split                                                                  | Test split                                                            |
> | Anomaly Detection    | Training split                                                                  | Test split                                                            |
> | Cooperative Game     | We gathered 1,000 episodes of trajectories using a crafted policy where the agent follows the optimal policy with a probability \(p = 0.7\) and a random policy otherwise. | We randomized the initial state and averaged the accumulative rewards over 30 episodes. |
>
> This structure ensures a clear separation of data used for rule extraction and performance evaluation, maintaining the fair of our results.
>
>
> ## Response to Question 3:
> > Given the reliance on deterministic predicates, how do you anticipate this approach adapting to real-world RAG scenarios requiring dynamic, knowledge-based decision-making?
>
> As noted in our response to Weakness 3,  it's important to highlight that our method also effectively handles non-deterministic scenarios:
> * We employ a dynamic knowledge retrieval mechanism from large language models (LLMs), leveraging semantic similarities with incoming inputs. This approach mirrors the flexibility found in Retrieval-Augmented Generation (RAG).
> * Additionally, LLMs are equipped with multiple logic rules and corresponding precision metrics, enabling them to judiciously choose the most applicable rules for each specific situation.
> * We have further validated the versatility of our approach through its successful application in a real-world industrial task, specifically in detecting Unauthorized Party Abuse (UPA). This demonstrates its efficacy in dynamic and knowledge-intensive environments.

---

> > ### Author Response · Authors · 2024-11-25
> >
> > Dear Reviewer fTJ7,
> >
> > As the discussion period draws to a close, we would like to confirm that our responses have thoroughly addressed the questions and concerns raised in your initial reviews. We are confident that we have effectively addressed both major and minor points in our replies. If there are any further questions or clarifications needed, we are eager to continue the discussion. If you find our responses satisfactory, we kindly request you to consider raising the score.
> >
> > Best regards,
> > The Authors

---

> > ### Author Response · Authors · 2024-12-03
> >
> > Dear Reviewer  fTJ7,
> >
> >
> > As the rebuttal period nears its end, we wanted to kindly follow up on our previous response. We sincerely hope the revisions and clarifications provided address your concerns and align with the criteria for an acceptable-level score.
> >
> >
> > If you have any additional comments or suggestions, we would be more than happy to address them before the discussion period concludes.
> >
> >
> > Thank you again for your thoughtful feedback and for considering our responses. We truly appreciate your time and effort.
> >
> >
> >
> > Best regards,
> >
> > The Authors

---

### Meta-Review · Area_Chair_c7Zy · 2024-12-21

**Metareview:**

The paper proposes a novel framework that distills offline data into logical rules using Monte Carlo Tree Search (MCTS), integrating these rules with LLMs to enhance reasoning across tasks. Strengths include its scalability, computational efficiency compared to methods like RAG, and its demonstrated effectiveness in diverse tasks (e.g., relation extraction, anomaly detection, and cooperative games). Reviewers appreciated its innovation in leveraging LLMs for rule-based reasoning and its empirical results showing improvement over baselines. However, weaknesses include unclear scoping on the method's generality, limited discussion on when rules are applicable, and insufficient clarity in some sections (e.g., predicate definitions, experimental details). The authors have conducted a satisfactory job on addressing these responses during the rebuttal. The paper has received borderline scores of 5,6,8, where the reviewer who gave 5 asked clarification questions and raised points on the inherent limitation of the method, such as the rule-based approach may be difficult to be applied in real-world tasks. This reviewer did not further participate in the discussion after the author responded. Personally I agree on this limitation but I think that is tolerable and the paper is interesting, thus I lean towards acceptance of this paper.

**Additional Comments On Reviewer Discussion:**

Reviewers highlighted strengths in the novelty of using LLMs to generate logical rules and its strong empirical results across tasks, but raised concerns about unclear scoping (when the method is applicable), insufficient experimental details, and clarity issues (e.g., predicate definitions, figure explanations). They also requested more fair comparisons with RAG and ICL. The authors addressed these by clarifying the method's scope, tasks, and predicate definitions, improving figure explanations, adding baselines for RAG and ICL, and revising experimental details (e.g., data splits, prompts). These responses satisfied most reviewers, with some raising scores and acknowledging the improvements. Overall I think this paper is novel and studies an interesting topic on finding rules with LLMs and augmenting generations with the rules.

---

### Decision · Program_Chairs · 2025-01-22

Accept (Poster)